# Interactions between precipitation, evapotranspiration and soil moisture-based indices to characterize drought with high-resolution remote sensing and land-surface model data.

Jaime Gaona[1,2], Pere Quintana-Seguí[2], María José Escorihuela[3], Aaron Boone[4], María Carmen Llasat [5]

[1]Instituto de Investigación en Agrobiotecnología, CIALE, Universidad de Salamanca, Villamayor-Salamanca, 37185, Spain.
[2]Hydrology and climate change, Observatori de l'Ebre (Universitat Ramon Llull - CSIC), Roquetes, 43520, Catalonia, Spain.
[3]IsardSAT, Parc Tecnologic Barcelona Activa, Barcelona, 08042, Catalonia, Spain.
[4]GMME/MOANA (Groupe de Météorologie à Moyenne Echelle), MOdélisation de l'Atmosphère Nuageuse et Analyse CNRM-GAME (URA CNRS & Météo-France), 42, Av. G. Coriolis, 31057, Toulouse Cedex 1, France.
[5]Applied physics department, Faculty of Physics, University of Barcelona, Barcelona, 08028, Catalonia, Spain.

*Correspondence to*: Jaime Gaona (jaimegaona@usal.es)

**Abstract.**

The Iberian Peninsula is prone to drought due to the high variability of the Mediterranean climate with severe consequences for drinking water supply, agriculture, hydropower, and ecosystem functioning. Because of the complexity and relevance of droughts

in this region, it is necessary to increase our understanding of the temporal interactions of precipitation, evapotranspiration and soil moisture that originate from drought within the Ebro basin, in northeast Spain, as the study region. Remote sensing and land-surface models provide high spatial and temporal resolution data to characterize evapotranspiration and soil moisture anomalies in detail. The increasing availability of these datasets has the potential to overcome the lack of in-situ observations of evapotranspiration and soil moisture. In this study, remote sensing data of evapotranspiration from MOD16A2 and soil moisture

data from SMOS1km as well as SURFEX-ISBA land-surface model data are used to calculate the EvapoTranspiration Deficit Index (ETDI) and the Soil Moisture Deficit Index (SMDI) for the period 2010-2017. The study compares the remote sensing time series of these ETDI and SMDI indices with the ones estimated using the land-surface model SURFEX-ISBA, including the Standardized Precipitation Index (SPI) computed at a weekly scale. The study focuses on the analysis of the time lags between the indices to identify the synchronicity and memory of the anomalies between precipitation, evapotranspiration and soil moisture.

Lag analysis results demonstrate the capabilities of the SPI, ETDI and SMDI drought indices computed at a weekly scale to inform about the mechanisms of drought propagation at distinct levels of the land-atmosphere system. Relevant feedback both for antecedent and subsequent conditions is identified, with a preeminent role of evapotranspiration in the link between rainfall and soil moisture. Both remote sensing and the land-surface model show capability to characterize drought events, with specific advantages and drawbacks of the remote sensing and land-surface model datasets. Results underline the value of analyzing drought

with dedicated indices, preferably at a weekly scale, to better identify the quick self-intensifying and mitigating mechanisms governing drought, which are relevant for drought monitoring in semi-arid areas.

## 1 Introduction

Drought is a major natural hazard for societies in semi-arid climates (Van Loon, 2015) which demands increasing levels of adaptation and resilience measures to guarantee water supply (Watts et al., 2012), particularly in water-stressed environments.

Rain-fed agriculture (Tigkas and Tsakiris, 2015), and even the enduring natural vegetation are very exposed to drought, especially under climate change, which has long-lasting implications for the local environment (Gudmundsson et al., 2014). Knowing that

complex interactions take place in the land-atmosphere system under drought, the traditional meteorological or hydrologic approach may overlook drought-relevant interactions between evapotranspiration and soil moisture (Teuling et al., 2013).

To track drought status and to analyze the interactions of the atmosphere-land system modern drought monitoring combines evapotranspiration, soil moisture and even vegetation anomalies in composite drought indices, such as the Objective Drought Indicator (OBDI) integrated in the U.S. Drought Monitor (Svoboda et al., 2002) or the Combined Drought Indicator of the European Drought Observatory (Sepulcre-Cantó et al, 2012). The use of this combined approach to monitoring drought is on the upward trend because even parsimonious composite drought indices like the probabilistic precipitation vegetation index (PPVI)
(Monteleone et al., 2020) outperform the capabilities of common indices to characterize drought. One of the major advantages of composite indices is that they facilitate the characterization of drought from multiple perspectives (e.g., Meteorological, Hydrological or Agricultural). Conversely, composite indices can be impractical to explore the mechanisms of drought, whose understanding may require focusing on key variables of the system. Unfortunately, evapotranspiration and soil moisture are still challenging to monitor compared to the meteorological, hydrological or vegetation variables currently regularly recorded. Despite
the relevance of these two variables in the recurrence of drought and heat waves (Zampieri et al., 2009; Dasari et al., 2014), even at short time scales (Teuling et al., 2018), relatively few studies have evaluated their anomalies due to the limited availability of data of sufficient spatial and temporal resolution.

Well-known drought indices such as the standardized precipitation index (SPI) (McKee, Doesken and Kleist, 1993) and the Palmer
drought severity index (PDSI) (Palmer, 1965), primarily defined on the monthly scale, can lack detail to identify short-term anomalies of temperature, wind or radiation originating "flash droughts" (Otkin et al., 2013). Rain-fed agriculture and natural vegetation are particularly sensitive to quickly evolving droughts in specific moments of the growing season (Saini and Westgate, 1999), which subsequently generates evapotranspiration and soil moisture anomalies of short and long-term impact (Jimenez et al., 2011). Recently, there is more interest in using drought indices with high temporal resolution for short-term drought monitoring,
such as the SPI and other indices at the weekly scale (Otkin et al., 2015). Indices with this short-term time scale include the weekly-scale evapotranspiration deficit index (ETDI) and the soil moisture deficit index (SMDI) (Narasimhan & Srinivasan, 2005). The ETDI and SMDI indices are variable-specific enabling full characterization of anomalies at specific levels of the land-atmosphere system. This is especially useful in the Mediterranean climates where not only rainfall anomalies originate drought (Vicente-Serrano et al., 2004).


This study focuses on the Ebro basin, which is an important Mediterranean river basin of the Iberian Peninsula (IP). In view of the increase in the frequency of drought events (Sousa et al., 2011) and the number of consecutive dry spells (Turco and Llasat, 2011) identified in the area, we can expect consequences in the long-term environmental state and the balance between water availability and demands. Furthermore, being placed in a semi-arid climate where most of the rainfall evaporates (68%, Table 15, of "Libro
blanco del agua" (MMA, 2000)), the Ebro basin represents an example of how important natural water demands are, particularly in the headwaters where reforestation decreases runoff (López-Moreno et al., 2014). Rainfed agriculture dominates the rest of the un-forested areas and represents the other big consumer of water in the basin. However, despite the relevance of rainfed agriculture its analysis is often overshadowed by irrigation, the biggest anthropogenic demand in the basin (Hoerling et al., 2012). Due to the importance of these water demands and others such as hydropower and energy, the Ebro River Basin Agency operates a dense
hydrologic monitoring network, but the lack of dedicated soil moisture and evapotranspiration monitoring jeopardizes drought

characterization (Seneviratne et al., 2010). Fortunately, the increasing availability of remote sensing (RS) products enables distributed, precise, and frequent monitoring of these coarsely observed variables (Martínez-Fernández et al., 2016).

Space agencies have released multiple RS products in the last decade facilitating the distributed analysis of drought (AghaKouchak et al., 2015). Optical spectrometry of the atmospheric (rainfall, temperature, water vapor) and surface (vegetation reflectance) variables have often been the basis for distributed characterization of drought indicators. Surface vegetation indices such as the widespread NDVI (Liu and Kogan, 1996) pioneered the application of RS data to assess the impacts of drought, but thereafter the increasing availability of RS data for multiple meteorological variables has increased its usage on drought indices (West et al., 2019). Currently, common indices like the SPI can rely on RS data (Sahoo et al, 2015), because integrating the increasing resolution of RS data into drought indicators enables short-term drought monitoring at least at the weekly scale (USDM, Svovoda et al., 2002; CDI, Sepulcre-Cantó et al., 2012; Monteleone et al., 2020). However, unlike precipitation, temperature and other directly observable and densely monitored meteorological variables, the measurement of evapotranspiration and soil moisture on the ground is still challenging and often costly or impractical at sufficient spatial resolution. Overcoming this gap is possible now thanks to the increasing availability of RS-based evapotranspiration databases such as the global dataset included in GLEAM (Miralles et al., 2011; Martens et al, 2017) or the soil moisture global database CCI (Dorigo et al., 2017). Despite the coarse spatial resolution of these global datasets, the recent developments in RS processing and downscaling improve their applicability at regional spatial scales and short-time scales (Wagner et al., 2007). Aiming to gain insight into drought mechanisms, the availability of high-resolution datasets focused on such relevant variables of the land-atmosphere facilitates the use of single-variable drought indices such as the SPI, ETDI and SMDI, which is advantageous to analyze the interactions between variables during droughts.

On this basis, there are soil moisture datasets of increasing high resolution available from the combination of passive microwave sensors such as those from SMOS and SMAP missions (Kerr et al., 2010; Entekhabi et al., 2010; respectively) and active microwave sensors such as ASCAT or Sentinel-1 (Bartalis et al., 2007; Hornacek et al., 2012; respectively). This is the case of the high-resolution soil moisture product SMOS1km (Merlin et al., 2013; Molero et al., 2016, Escorihuela and Quintana-Seguí., 2016; Escorihuela et al., 2018) which has been tested in the area and shown to outperform ASCAT and ASMR-E due to its lack of roughness and vegetation effects. SMAP and Sentinel-1 options are of similar resolution to SMOS1km, accurate in the study area (Dari et al., 2021), but of much shorter series length, and consequently not selected. Similarly, high-resolution RS evapotranspiration products such as the MOD16A2 (Mu et al., 2013) used in this study are currently available. Therefore, it is worth exploring the capabilities and limitations of high-resolution RS evapotranspiration data for drought monitoring at the regional scale. High-resolution RS data are best suitable for analysis at the basin scale where the resolution of alternative reanalysis or modelled datasets such as ERA5-Land (Muñoz-Sabater et al., 2021) or LISFLOOD (Van der Knijff et al., 2008) GLEAMv3 (Miralles et al., 2011; Martens et al, 2017) lack detail. To date, relatively few works have used satellite high-resolution data for drought analysis in the IP (Vicente-Serrano, 2006; Scaini et al, 2015, Martínez-Fernández et al., 2016; Sánchez et al., 2016; Ribeiro et al., 2019), especially at the spatial and temporal resolution of this study (Pablos et al., 2017).

Another source of high-temporal and spatial resolution data is land-surface models (LSM). Used in atmospheric models to simulate the interactions between soil, vegetation and the atmosphere, LSMs represent a suitable alternative to RS to evaluate the surface water and energy balances at regional to local scales. LSMs initiated their development with one-layer models such as the TOPUP (Schultz et al., 1998) or the PROMET (Mauser and Schadlich, 1998). Avissar and Pielke (1989) inaugurated the mosaic approach, applying just one-layer models to the different fractions of land-use type. One of the mosaic models able to distinguish between

soil evaporation and transpiration is the Météo-France developed model SURFEX (Masson et al., 2013), which fed by the atmospheric analysis SAFRAN (Durand et al., 1999) uses the ISBA scheme for natural surfaces (Noilhan and Mahfouf, 1996). SURFEX has been improved to study the continental water cycle in applications such as SIM and SIM2 (Habets et al., 2008; Le Moigne et al., 2020), often in combination with the hydrologic model MODCOU (Ledoux et al., 1989). The modelling chain called SASER (SAFRAN-SURFEX-Eaudyssée-RAPID) used in this study has been applied to Spain before (Barella-Ortiz and Quintana-Seguí., 2019; Quintana-Seguí et al, 2020). This LSM provides the precipitation required for SPI and the evapotranspiration and soil moisture necessary to generate LSM-based ETDI and SMDI series comparable to the ones generated using RS data. Despite the limitations of this LSM when applied as an offline model, it has been validated and reported to provide useful evaluations of water resources in the study area (Escorihuela and Quitana-Seguí, 2016; Barella-Ortiz and Quintana-Seguí, 2019) and nearby Portugal and France (Nogueira et al., 2020; Le Moigne et al., 2020).

This study aims at evaluating the suitability of high-resolution RS (SMOS1km and MOD16A2) and LSM (SURFEX-ISBA) data for generating rainfall (SPI), soil moisture (SMDI) and evapotranspiration (ETDI) drought (single-variable) indices to better understand the mechanisms behind the temporal evolution of drought in semi-arid climates. The comparison of RS and LSM results is a main aim of the study to detect the factors impacting drought indices based on RS and LSM data. The study further evaluates the advantage of the barely explored weekly temporal scale to capture the short-term anomalies of evaporation and soil moisture decisive for drought in semi-arid areas. The study has an agricultural scope focused on drought in rain-fed environments given its importance on the land-atmospheric feedbacks (Herrera-Estrada et al., 2017) and the regional socioeconomic sustainability.

## 2 Study area

The study area is the Ebro basin, located in the northeast of the Iberian Peninsula (IP). Placed in between Atlantic and Mediterranean climatic influences, the vast area (85534 km2) of the basin (Fig. 1a) has a complex topography (Fig. 1c) which defines a wide range of climatic conditions (Fig. 1d) of distinct spatial and temporal patterns of precipitation, evapotranspiration and soil moisture. The northern border has humid cool climates typical of the Atlantic-exposed Cantabric coastline, while the southeast border enjoys a warm Mediterranean climate. The southwest and north-eastern border are dominated by the Iberian and Pyrenees mountains which together with the Cantabric and Mediterranean ranges restrict the oceanic influence on the central part of the basin. Soil types (e.g. gypsum, limestones) intensify the aridity of certain areas of the basin (Fig. 1b). The combination of semi-arid climatic conditions and unfavorable soil types to vegetation development determine extreme regimes of rainfall, soil moisture and evapotranspiration, prone to drought. The basin is densely populated and supplies a wide range of water demands, especially for agriculture and energy. The vast network of irrigated areas, located mostly in the arid central depression, is vulnerable to hydrological drought risks.

## 3 Data

### 3.1 Land-Surface Model Data

SURFEX, the land-surface modelling platform originally developed and currently maintained by Météo-France (Mason et al., 2013; Le Moigne, 2020), has been chosen to perform the LSM model simulation used in this study. Simulations for the IP and

Balearic Islands developed within the HUMID project have 5 x 5 km spatial resolution and use the forcing provided by the Iberian application (Quintana-Seguí et al., 2008; 2016; 2017) of the SAFRAN meteorological analysis system (Durand et al, 1999). This is a modelling chain whose offline mode operates using the atmospheric forcing of SAFRAN to feed the LSM SURFEX-ISBA and simulate even the hydrology with Eaudyssée-RAPID (David et al. 2011).

ISBA (Noilhan and Mahouf, 1996) is the SURFEX module in charge of simulating natural surfaces. There are different versions of ISBA. In this study, we have used the diffusion version (ISBA-DIF; Boone, 1999; Decharme et al., 2011), which performs better in the study area than the simpler 3-layer force restore version (Quintana-Seguí et al., 2020). In this version of ISBA, the LAI has a prescribed annual cycle (constant every year), which may limit the ability of the model to reproduce the long-term effects of

drought on vegetation. The model simulates the soil column, but it is unable to simulate groundwater, which despite its impact on soil moisture memory is not very relevant in the Ebro basin. SURFEX-ISBA requires additional physiographic information that is incorporated from the ECOCLIMAP II land cover database (Faroux et al., 2013) which includes topographic, soil and land cover information at high-resolution.

The available SAFRAN forcing data allows us to simulate the period 1979-2017, but the period used for this study is restricted by the relatively short length of the RS SMOS data (2010-present) compared to the model. To ensure the comparability of RS-based and LSM-based drought indices, the study period is 2010-2017, for which both RS and LSM data are available. To ensure that the RS and the LSM soil moisture are comparable, we have averaged three first soil layers of the model according to their discretization in the first 5 cm of the soil (1, 3 and 10 cm of depth respectively). The simulation is performed using a regular 5 x 5 km resolution

grid based on a custom Lambert Conical Conformal projection.

### 3.2 Remote Sensing Data

### 3.2.2 Evapotranspiration

To evaluate evapotranspiration, barely measured on the ground and not directly measurable from space, we adopt a product based on multiple evaporation-related variables observed by MODIS (Moderate Resolution Imaging Spectroradiometer on NASA's Terra

satellite): the MOD16A2 dataset. This is a level 4 product providing 8-day evapotranspiration (ET) and potential evapotranspiration (PET) based on daily meteorological forcing and 8-day RS data of vegetation dynamics from MODIS (Mu et al., 2013). The datasets of MOD16A2 are published in a sinusoidal projection at a resolution of 500 m (Running et al., 2017). In this study, we have re-projected and interpolated all RS products to the same 5 x 5 km grid that the LSM simulations use. After the re-gridding step, the temporal step of the datasets is rearranged from the original 8-day accumulation period to a 7-day accumulation period which is more suitable for weekly analysis. Values are linearly weighted depending on their contribution to each week-year for the

52 weeks of a year. We also calculate the monthly means of the evapotranspiration product to evaluate the impact of the time resolution on drought recognition. The formatting of MOD16A2 datasets requires the evaluation of the quality control flags (ET_QC), given that areas of the Pyrenees show missing data. Only the classes classified as good and optimal in the ET_QC flags are accepted as data for our study.

### 3.2.3 Surface Soil Moisture

Because of the relatively few years of data currently available from the Soil Moisture Active Passive mission SMAP (Entekhabi et al., 2010), the study adopts SMOS data (Kerr et al., 2010), in particular, the high-resolution SMOS1km dataset (Merlin et al.,

2013). This dataset downscales the original coarse resolution SMOS data using the Disaggregation based on Physical And Theoretical scale Change algorithm DISPATCH (Merlin et al., 2012) / C4DIS (Molero et al., 2016) algorithm. The algorithm enables the downscaling of the 40 km resolution of the SMOS soil moisture data available from 2010 into 1 km resolution using two products at 1 km resolution from MODIS, the NDVI and LST, and an elevation map at the same resolution. Precisely because the scale of interest to study relevant interactions to droughts is the weekly scale, the data is primarily used on a weekly scale. The spatial scale of interest for the study is that of a regular 5 x 5 km resolution grid which takes advantage of the high resolution of SMOS1km. The 2010-2017 dataset presents frequent gaps in the mountainous areas of the Pyrenees. In order to fill the gaps, we apply temporal interpolation pixel by pixel considering a maximum period for temporal interpolation of two weeks. These data are also spatially aggregated and reprojected to the same grid as the LSM model.

## 4 Methods

### 4.1 Drought indices

Drought indices allow quantifying several aspects of drought, like the magnitude and duration, and may also focus on particular variables depending on the scope of interest (i.e., precipitation, soil moisture, aridity…etc.). In view of the convenience to combine several single-variable indicators to describe most of the drought mechanisms and our focus on rain-fed environments, we adopt the use of the Standardized Precipitation Index (SPI) (McKee et al., 1993), the Evapotranspiration Deficit Index (ETDI) and the Soil Moisture Deficit Index (SMDI) (Narasimhan and Srinivasan, 2005). Using these three indices, the study aims to investigate the interaction between the two main water fluxes (rainfall and evapotranspiration) and the main storage (soil moisture) involved in the water balance of the land-atmosphere system. The aggregation periods of the SPI index inform about the different responding times of rainfall, soil moisture, streamflow, and groundwater anomalies. By evaluating the evolution of these indices along with their interactions, this study aims to characterize drought mechanisms in the Ebro basin. In this study, the indices have been computed using gridded datasets, thus, generating a time series for each grid point.

#### 4.1.1 SPI

The Standardized Precipitation Index (SPI) (McKee et al. 1993) is an index of precipitation anomalies, which is calculated by transforming the accumulated precipitation from its original distribution (usually gamma or Pearson Type III) to the normal distribution with zero mean and unit standard deviation. As a result, we obtain a time series that shows, for each time step, the departure from the expected value in terms of standard deviations. The calculation of the index is usually done based on monthly time series of rainfall, aggregated over multiple accumulation periods, typically at 3, 6, 12 months. However, the SPI can also be calculated on a weekly basis, provided that the accumulation periods are at least 4 weeks (1 month). In this way, this study uses the notation SPIm-*i* to denote the SPI at a monthly scale with an accumulation period of *i* months and the SPIw-*i* to denote the weekly SPI with an accumulation period of *i* weeks. Using SAFRAN data, we adopt the non-parametric methodology proposed by Farahmand and Aghakouchak (2015) to calculate the SPI on a monthly and weekly basis using the multi-month accumulation of 1, 3, 6 and 12 months suitable for rainfall, soil moisture, streamflow and groundwater evaluation.

#### 4.1.2 ETDI

The second index incorporated into the analysis is the EvapoTranspiration Deficit Index (ETDI) defined by Narasimhan and Srinivasan (2005). The first step for calculating this index implies defining the water stress ratio (WS) for each week, which is the

difference between potential evapotranspiration (PET) and actual evapotranspiration (AET) divided by PET. Then the Water Stress Anomaly (WSA) is computed as follows:

$$WSA_{i,j} = \frac{MWS_j - WS_{i,j}}{MWS_j - \min WS_j} x100, \quad if\ WS_{i,j} \leq MWS_j$$

$$WSA_{i,j} = \frac{MWS_j - WS_{i,j}}{\max WS_j - MWS_j} x100, \quad if\ WS_{i,j} > MWS_j \tag{1}$$

where $j$ denotes the week of the year ($1 <= j <= 52$) and $i$ denotes the year. $WS_{i,j}$ is the water stress of the week $j$ of the year $i$. $MWS_j$ is the median WS for the week $j$ of the year, $minWS_j$ corresponds to the minimum and $maxWS_j$ to the maximum. This process removes the seasonality of the time series. WSA ranges from -100 (maximum water stress) to 100 (minimum water stress). The WSA is accumulated over time to define the ETDI as in the following equation:

$$ETDI_j = 0.5\ ETDI_{j-1} + \frac{WSA_j}{50} \tag{2}$$

To define a range between -4 and 4 for the index, ETDI of value -4 must correspond to WSA of value -100 and ETDI of value 4 to WSA of value 100. This range adjustment determines the coefficients 0.5 and the divisor 50 of Eq. (2). In this way, ETDI becomes a non-seasonal index suitable for comparing time series of diverse climatic characteristics. Monthly values of the ETDI are calculated by computing the average of the weekly values of the corresponding month. In this study, we calculate the ETDI using potential and actual evapotranspiration provided by MOD16A2. We have also calculated it using the SURFEX-ISBA simulated AET and PET. By default, SURFEX-ISBA does not calculate potential evapotranspiration. In order to do so, we have modified the source code to set soil moisture permanently at field capacity and have run a simulation with this modification. The resulting evapotranspiration corresponds to the potential one.

**4.1.3 SMDI**

The third index incorporated into the analysis is the Soil Moisture Deficit Index (SMDI) also defined by Narasimhan and Srinivasan (2005). The sequence to calculate this index follows the same procedure as the ETDI. We first calculate a weekly soil moisture deficit (SD) as follows:

$$SD_{i,j} = \frac{SW_{i,j} - MSW_j}{MSW_j - minSW_j} x100, \quad if\ SW_{i,j} \leq MSW_j$$

$$SD_{i,j} = \frac{SW_{i,j} - MSW_j}{maxSW_j - MSW_j} x100, \quad if\ SW_{i,j} > MSW_j \tag{3}$$

Then, the time series of SMDI is generated accumulating SD:

$$SMDI_j = 0.5\ SMDI_{j-1} + \frac{SD_j}{50} \tag{4}$$

SMDI also ranges between -4 and 4, respectively corresponding to extremely dry and very wet soil moisture conditions. The SMDI, similarly to the ETDI, becomes a non-seasonal index able to compare time series of diverse climatic characteristics at different soil depths. In this study, we have calculated the SMDI using SMOS1km surface soil moisture data and SURFEX-ISBA simulated surface soil moisture data. In the case of SURFEX-ISBA, we have calculated the weighted averages of the first two layers of the soil, which corresponds to the first 5 cm of the soil.

**4.1.4 Temporal consistency of drought indices calculated based on relatively short RS and LSM data series**

Given the relatively short availability of data for the calculation of ETDI and SMDI series, which depend on the maximum, minimum and median values of the available series, we conducted a sensitivity analysis of the indices in reference to the length of the series and the subset of spatial data. Results shown in Table S1 illustrate the relatively low impact of the length of the series thanks to the high spatial resolution of the dataset. The shortening of the series by half or a quarter barely alters the ETDI series compared to the case of using the full temporal length. The subset of the dataset to a fraction of its spatial resolution increasingly

impacts the robustness of ETDI and SMDI series. Therefore, the high-resolution spatial and temporal datasets such as the RS and LSM used for this study support the consistency of drought indices even when data availability remains under a decade long.

**4.2 Analysis of interaction between the indices**

**4.2.1 Correlation between the indices**

To evaluate the similarity between the series of the drought indices (SPI, ETDI and SMDI) we use a variant of the procedure applied by Barella-Ortiz and Quintana-Seguí (2019) and Quintana-Seguí et al. (2020), based on Barker et al. (2015). The method consists of computing the r Pearson correlation between each pair of series of these three drought indices (e.g. SPIw-i and ETDIw and SMDIw); where $i$ is the accumulation period, which varies from 4 weeks (1 month) to 52 weeks (1 year). For this evaluation and the following lag-analysis, we adopt the r Pearson coefficient because the time it consumes to process our RS and LSM datasets

is an order of magnitude lower (e.g. weeks) than the time required using r Spearman (e.g. months). To further support the use of r Pearson despite the concerns about the non-normality of SMDI and ETDI distributions, we conducted a similarity test of the series of lags between indices obtained with r Pearson and r Spearman. Similarity, tests indicate r Pearson and r Spearman correlate generally over $r = 0.9$ for RS data and moderately lower for LSM data while do not differ significantly in the timing characteristics of the series (Fig. S2). Therefore, we can consider r Pearson suitable and sufficiently accurate for the approach and focus of the

study.

**4.2.2 Temporal lag-analysis of the indices:**

    Following the correlation analysis between the series, we perform a lag analysis of the correlation of the pairs of drought indices at a weekly scale, introducing lags from -104 weeks to +104 weeks. We compare the ETDIw and SMDIw with the SPIw-i (being

the period of accumulation for the SPIw-i= 4, 13, 26 and 52 weeks, equivalent to the SPIm-1, SPIm-3, SPIm-6, SPIm-12 months), as well as the ETDIw with the SMDIw. The purpose of this analysis is to diagnose the reciprocity and memory in the interaction between rainfall and evapotranspiration, and rainfall and surface soil moisture. The relative abundance of positive over negative lags (and vice versa) provides information about the asymmetry of the interaction between the indices (precedence and delay). Negative lags refer to leading times of ETDIw and SMDIw in respect of SPIw-i (e.g. lag - 104, left side of the time bar) while

positive ones (e.g. lag +104) represent lag times when SPIw-i precedes ETDIw and SMDIw (e.g. lag +104, right side of the time bar). The number of consecutive weeks of positive or negative lags of the SPIw-i can inform about the memory of the interactions. For each time lag, it is indicated the percentage of the basin affected by non-significant / significant correlations (Grey / colored scales of bars in Figs. 3-7). Positive /negative correlations indicate a direct / indirect relationship (red / blue bars in Figs. 3-7).

# 5 Results

## 5.1 Correlation between indices: Monthly scale in comparison to the weekly scale

The first two aims of the study are to evaluate the suitability of SPI, ETDI and SMDI indices to characterize the main anomalies in water exchanges of the land-atmosphere system and to evaluate the suitability of adopting the weekly scale for the analysis of drought indices compared to the use of the monthly scale. Regarding the first, results shown in Fig. 2 indicate a general agreement of the SPI, ETDI and SMDI indices (either computed at a monthly or weekly scale) on the major events of dry and wet anomalies of the period 2010-2017. The dry period of 2011-2012 and the wet period from the end of 2012 to 2015 were properly depicted. However, we identified differences, especially in the case of the SMDI. This index tends to show a generally lower variability than other indices when calculated with the LSM. RS results of SMDI differ from the other indices during the start of 2010 due to the uncertainties during the test period of the SMOS mission. The left column of Fig. 2 shows the monthly SPIm-i and the monthly averaged ETDIm and SMDIm. Correlations between SPI-i and ETDI / SMDI are calculated on the monthly (Table 1, left columns) and on weekly scales (right columns) for both RS and LSM data (where i= 4, 13, 26 and 52 weeks of aggregation, equivalent to 1, 3, 6 and 12 months). We test the significance of the correlations (p-value < 0.05) between indices for two subsets: the entire period (2010-2017) and a subset of dry periods (i.e., when SPI<0, ETDI<0, SMDI<0). Then, we compare the observed (RS) and simulated (LSM) estimates of the indices to explore differences between data sources. RS and LSM ETDIm and SMDIm indices are moderately correlated (barely over 0.5, significant). Table 1 reports a value of r=0.58 (significant) between ETDIm RS and SMDIm RS which is significantly higher than the r=0.32 between ETDIm LSM and SMDIm LSM. In general, despite the resemblance of RS and LSM series of ETDI and SMDI shown in Fig. 2, these two series differ. There is a higher agreement between the RS and the LSM estimates of ETDIm (r = 0.77, significant) than between RS and LSM ones of SMDIm (r = 0.27).

Table 1 also reveals differences in the moderate correlation of the SPIm-i with the ETDI and the SMDI of different temporal aggregations (i=1,3,6,12 months of SPIm-i accumulation) as well as between their RS and LSM versions. Correlation between SPIm and ETDIm RS increases with the aggregation period of the SPIm (r = 0.39, 0.62, 0.72, 0.81 respectively for SPIm-1, SPIm-3, SPIm-6, SPIm-12) while correlations of SPIm with ETDI LSM peak at SPIm-3 (r=0.8) and remain high for SPIm-6 (r=0.78) and SPIm-12 (r=0.71). The correlations between SPIm-i and SMDI RS show moderate correlation ranging from the SPIm-1 (r=0.42) to the maximum value of the SPIm-12 (r=0.63). SMDIm LSM exhibits a decreasing correlation pattern with the increasing aggregation of the SPIm from SPIm-3 to SPIm-12 (r=0.45, 0.31, 0.3, 0.24 respectively for SPIm-1, SPIm-3, SPIm-6 and SPIm-12). Remarkably, correlations of LSM SPIm - SMDIm are lower than their RS pairs, which suggests data uncertainties in RS and LSM, as reported by Barella-Ortiz and Quintana-Seguí (2019) and by Quintana-Seguí et al. (2020)).

The monthly correlation analysis is additionally conducted for the subsets of dry periods (those with negative signs of the SPIm, ETDIm and SMDIm). Using the dry subset instead of the entire period primarily decreases all correlations. Compared to the correlations of the entire period, the RS ETDIm – SMDIm values decrease a bit while LSM ones increase a bit. Also the SPIm-i – ETDIm / SMDIm correlations decline noticeably. Despite the loss of correlation and significance with the dry subset, they experience an increase in correlations with the increase of the aggregation periods, equally to the case of the entire period series.

Fig. 2 provides an overview of the effect of adopting the weekly scale (right column) instead of the monthly scale (left column). The weekly scale substantially improves the temporal resolution of the plots. Subplots of SPI-i (a to d), ETDI (e) and SMDI (f) show how the weekly scale accurately reproduces the magnitude, tendency, and duration of the monthly-scale anomalies, while

the increase in temporal resolution additionally captures quick changes. Graphically, we noticed that the aggregation period applied to the SPI counteracts the gain of resolution of the weekly scale, which often prevents users from adopting the weekly scale instead of the monthly scale. Conversely, Fig. 2 $e_2$ and $f_2$ illustrate the strong increase in the resolution of the ETDI and SMDI resolution on the weekly scale.

The right columns of Table 1 indicate the correlations between indices on a weekly scale compared to those on a monthly scale, (the 'w' sub-index of ETDIw / SMDIw / SPIw denotes the weekly scale). There is an overall decrease in correlations at the weekly scale compared to the monthly scale, accentuated by the increasing period of aggregation (from SPIw-3 to SPIw-12). Correlations increase a bit compared to the monthly scale at the lowest period of aggregation of SPIw-i, especially for the ETDI (from r=0.39 to 0.57 / r=0.51 to 0.68 in SPIw-1 - SMDIw RS / LSM). For both RS and LSM data, the weekly scale lowers the correlation values of the SPIw-i with the SMDIw more than those with the ETDIw. Within the dry-period subset, all the correlations decrease too. The weekly scale lowers correlations due to the increase of variability of the weekly time series compared to the monthly ones but enables capturing the increasing complexity of interactions at shorter time scales. Therefore, since quick shifts (Fig. 2 $e_1$ and $f_1$) are of great interest for drought analysis, we consider the weekly scale as the most convenient for the lag analysis of the next section. This decision benefits comparing drought indices at their highest temporal definition, which for the ETDI and SMDI indices is natively the weekly scale and for the SPI can be easily adopted.

**5.2 Temporal lag analysis:**

The analysis of correlations between pairs of temporally lagged time series of drought indices provides valuable insights into the interactions between indices (e.g., in terms of reciprocity) as indicators of synchronicity of one variable with respect to the other. The analysis addresses the characterization of the interactions between rainfall, evapotranspiration and soil moisture while also aims to examine whether the land-atmosphere exchange of semi-arid areas under drought events is a quickly evolving or rather inertial system. Plots of the fraction of the area affected by each correlation level (from -1 to 1 in steps of r=0.2) for each lag period are shown in Figures 3 - 7. It is anticipated that the three evaluated indices agree the most in a narrow range around lag = 0 (no lag) and that their correlation fades away progressively for the increasing lag periods beyond the scale of the propagation of drought. Figures 3 to 6 in which the SPIw-i interacts with ETDI and SMDI illustrate the range of aggregation periods of SPIw (i.e., from 4 weeks of aggregation SPIw-4 to SPIw-52) along which the interpretation of the interactions between drought indices becomes most altered. The range of aggregation periods aims to open discussion about the optimal time scale to analyze the interacting drought processes. In general, the upper two subplots of Figs. 3 to 6 identify the admissible aggregation periods to interpret the clusters of interaction (SPIw-4 and SPIw-13, equivalent to SPI-1 and SPI-3 on a monthly scale) while the bottom two subplots (SPIw-26 and SPIw-52, equivalent to SPI-6 and SPI-12 at monthly scale) depict the increasingly merged clusters of interaction due to excessively long aggregation periods of the SPI.

**5.2.1 Lag analysis SPI-ETDI**

**RS**

The lag analysis of SPIw – ETDIw and SPIw – SMDIw shown in Fig. 3 - 6, aims at diagnosing the reciprocity, synchronicity and memory in the interaction between rainfall and evapotranspiration, and rainfall and surface soil moisture. Each subplot of Fig. 3 to 6 shows the correlation between the ETDIw and SMDIw indices with the SPIw-i index calculated for different aggregation periods (-i) at the week scale: the SPIw-4, SPIw-13, SPIw-26 and SPIw-52. Negative lags refer to leading times of ETDIw and

SMDIw in respect of SPIw-i (e.g., lag - 104, left side of the time bar) while positive ones (e.g. lag +104) represent lag times when SPIw-i precedes ETDIw and SMDIw (e.g., lag +104, right side of the time bar).

In the case of the ETDIw – SPIw-i analysis based on the RS data, there is a remarkable cluster of positive correlations in the short term (indicated with the tag 'ST$_1$' (Short-Term) over the subplots of Fig. 3). This 'ST$_1$' cluster show relevant fractions of the basin affected by significant moderate (0.4-0.6) values of correlation, particularly in the first weeks of the positive range of lags (from lag 0 to +4) (Fig. 3). The cluster lasts more with the increasing period of aggregation (Fig. 3 a) to d)) eventually becoming merged with the mid-term clusters. In view of subplots Fig. 3 a) to d), the 'ST$_1$' cluster extends from lag -13 to 4 in the case of SPIw-4, from lag -13 to + 13 in the case of SPIw-13, and once merged with 'MT$_1$' from -26 to +26 in the case of SPIw-26 and from -26 to +39 for the SPIw-52. The mid-term cluster 'MT$_1$', originally indicating a period of correlation of ETDIw preceding SPIw-i from 4 to 13 weeks of aggregation, displays moderate to low but significant values of correlation (from 0.2 to 0.4) from lag -36 to -13. The merging of the cluster 'ST$_1$' and 'MT$_1$' decreases the asymmetry defining the leading role of ETDIw on SPIw-i. The initially asymmetric interaction of ETDIw – SPIw-i that is mostly located in the negative range of lags ('ST$_1$' and 'MT$_1$' shown between lag -30 and +10 for the SPIw-4 and SPI-13 cases) propagates and dampens towards the positive range of lags with the increasing aggregation. The dampening eventually shifts the interaction of ETDIw to SPIw-i from preceding to delayed (at 26 and 52 weeks of aggregation, the clusters of significant positive correlations are mostly within the positive range of lags, especially in Fig. 3 d). The loss of asymmetry due to the increasing aggregation translates into an increase in the duration of the cluster as well as of the fraction of the basin significantly affected by correlations. Both effects may indicate the inconvenience of adopting long aggregation periods that alter the interaction magnitude and timing. There is an additional cluster of positive correlations 'LT$_1$' past the year and half (lags +78 to +104), particularly noticeable at SPIw-13 and -26. Blue bars in Fig. 3 indicate negative correlations for the relationship ETDIw – SPIw-i dominating the long-term between evapotranspiration and rainfall anomalies. There is a couple of clusters around lag +42 (tagged with 'LT$_2$') and around lag +104 ('LT$_4$'), slightly significant, that similarly to the positive correlations, increased in duration and magnitude with the increase of the aggregation period of the SPI, particularly for SPIw-26 and 52 (Fig. 3c - d).

**LSM**

Results from the LSM SURFEX-ISBA (Fig. 4) show a less lasting and more concentrated cluster of significant positive correlations around lag 0 ('ST$_1$' tagged in Fig. 4) than those observed in the RS results (Fig. 3). This result implies there is more synchronicity between SPI and ETDI in LSM data than in the RS data. Furthermore, the LSM provides higher magnitudes of the significant positive correlations of ETDIw – SPIw-i of ST$_1$ than RS results. Similarly to RS data, the duration of the highly correlated period 'ST$_1$' extends with the increasing aggregation of the SPIw-i (Fig. 4 a to d), eventually causing the merge of clusters 'ST$_1$' and 'MT$_1$'. The initial asymmetry of the positive correlations towards the negative range of lags is due to the cluster 'MT$_1$' placed around lag -26 and cluster 'LT$_1$' (Fig. 4 a and b). In the SPIw-26 and SPIw-52 cases (Fig. 4 c and d) 'LT$_1$' disappears, and the 'ST$_1$' merges with 'MT$_1$', artificially shifting the initial asymmetry dominating the negative range of lags towards the positive range. Thus, the precedence of ETDIw with SPIw-i prevalent in the period of aggregation of 4 and 13 weeks may look like the precedence of SPIw-i with ETDIw when long aggregation periods such as the ones in SPIw-26 and -52 are applied. The initial asymmetry of the cluster 'ST$_1$' – 'MT$_1$' is lower in LSM results (Fig. 4) than in RS results (Fig. 3) due to the lower magnitude of 'MT$_1$'. The additional cluster of positive correlations 'LT$_3$' past the year and a half range of positive lags (lags +78 to +104) in LSM results (Fig. 4a to c) concurs with that of RS results (Fig. 3a to c). LSM results (Fig. 4) show a few more clusters of negative correlations ETDIw – SPIw-i but of lower magnitude than those of RS ones (Fig. 3). These significant clusters merge with the

increasing aggregation period 'LT$_4$ – LT$_5$', which agrees with 'LT$_2$' shown in RS at the 26- and 52-weeks period of aggregation of SPIw-i. LSM results further include a cluster of significant negative correlations values in the lead time range (cluster 'LT$_2$') that is absent in RS results (Fig. 4 vs. Fig. 3). The agreement between LSM and RS results confirms the asymmetrical interaction between ETDIw and SPIw-i. The asymmetry suggests a prevalence of the precedence of positive correlations of ETDIw with SPIw-i in the short term (from the month to the seasonal scale) while points to some delayed response of ETDIw to SPIw-i in the negative correlations in the mid to long-term (from seasonal to interannual scale). The LSM results tend to amplify the magnitude and the area affected by positive correlations compared to the RS dataset.

### 5.2.2 Lag analysis SPI-SMDI

**RS**

The interaction of SPIw-i with SMDIw for the RS dataset is not as strong as it was with ETDIw (Fig 5 and 6 vs. Fig. 3 and 4). Both the correlation values and the significant fraction of the basin affected by them are lower than in the case of the ETDIw. Similarly to the ETDIw, the significant positive correlations around lag 0 ('ST$_1$') are asymmetrical. Because of this, the SMDIw tends to experience the effect of the preceding conditions of the SPIw-i less than influencing those of the SPIw-i. The increasing period of aggregation of the SPIw-13, -26 and -52 widens and lags the short-term influence of the SPIw-i on the SMDIw ('ST$_1$') into a short to mid-term influence ('ST$_1$-MT$_1$'). This widened cluster of positive correlations stays almost entirely in the positive range of lags, which differs from that of the ETDIw where the 'ST$_1$-MT$_1$' cluster extended both in positive and negative ranges of the lags (Fig. 5 compared to Fig. 3 to 4). The leading range of lags (from lag -104 to lag 0) does not show relevant clusters of significant correlation at all, neither in the mid nor in the long term. Negative correlations of the SMDIw – SPIw-i interaction only occasionally stand up at a small cluster ('LT$_1$', Fig. 5 b - d) in the positive range of lags (when SPIw-i precedes SMDIw). The smoothing effect of the aggregation period of SPIw-i increasingly alters the initial bias of interactions towards the leading influence of SMDI on SPIw, as well as delays most clusters similarly to the case of with ETDIw.

**LSM**

The LSM results for the SMDIw – SPIw-i relationship depicts strongly dampened patterns of significant correlation between SMDIw and the SPIw-i compared to those of RS SMDIw – SPIw-i or ETDIw – SPIw-i. Only the 'ST$_1$'cluster is noticeable around lag 0. The increase in the aggregation period does not favor its permanence as a relevant cluster beyond the SPIw-4. The cause can be the generalized low values of both non-significant and significant correlations obtained for these estimates of LSM SURFEX-ISBA which may indicate the difficulties of the LSM to describe the response of the surface soil moisture (the SMDI index) to the atmospheric forcing (the SPI index) that we saw in the RS dataset. The periods identified of short-term, mid-term and long-term influence of one SPIw-i in the SMDIw such as 'SP$_1$', 'MT$_1$', 'LT$_1$' cannot be recognized in Fig. 6 compared to Fig. 5. Therefore, the LSM results of the SPIw-i – SMDIw relationship strongly differ from the ones obtained from RS data and is a matter of debate in the discussion section.

### 5.2.3 Lag analysis ETDI-SMDI

**RS**

The remaining interaction in this analysis is the ETDI – SMDI. The results show a less asymmetric relationship between the ETDI and the SMDI compared to the ones between SPIw with ETDI and SMDI. The significant moderate positive correlation values (red bars in Fig. 7a) between lag 0 and +13 and from lag -10 to 0 indicate that the influence of the ETDI on the SMDI lasts longer

than the one of the SMDI on the ETDI. The magnitude, though, expresses that SMDI moderately impacts the short-term conditions of the ETDI for about a month in comparison to the sole week the ETDI affects moderately those of SMDI. However, the highest correlations occur for the lag -1 when SMDI precedes ETDI in one week. Negative clusters 'MT$_1$, MT$_2$' at +/- 39 weeks suggest that the interaction between the indices goes beyond the seasonal scale commented above. However, the significance of all these mid to long-term clusters remains low.

**LSM**

The results of ETDI – SMDI interaction based on LSM data show less evident periods of interaction between the indices compared to the RS results. The expected strong correlation around lag 0 is largely diminished. The strongest cluster appears from lag -21 to -39 'MT$_1$' (Fig. 7b) when SMDI precedes ETDI. No notable negative clusters can be identified. Apart from the lack of agreement on the symmetry of the interaction between LSM and RS results (Fig. 7b vs. 7a), the notable cluster 'ST$_1$' in RS results is less relevant in LSM results. Given the disparity between LSM and RS results, we rise concerns about the accuracy of offline LSM simulations compared to the RS results, addressing them for discussion in the next section.

**6 Discussion**

Results require careful discussion regarding three main aspects: firstly, the effect of adopting the weekly scale for drought indices and analyses, secondly the meaning behind the complex interactions between drought indices and thirdly the comparison of RS and LSM as tools for high-resolution monitoring of drought. All comments refer to the results on a weekly scale.

**6.1 Scales for drought monitoring in semiarid environments**

Analyzed the differences in correlations between indices at monthly and weekly scales (Figure 2), we support the necessity of adopting the weekly scale to study lags. The monthly scale preferred for drought assessment from a hydrological perspective may overlook the quick response of the land-atmosphere interactions. The clusters of moderate to high correlation between indices mostly occur within the first month preceding or following an anomaly (Figs. 3-7), particularly in the short to very short-term. High correlations tend to peak and plunge in the interval of a few weeks. This short-term response recommends the use of the weekly scale and aggregation periods below the seasonal scale, such as SPIw-13 (equivalent to SPI3), to evaluate the delay or precedence between indices.

Our results showing soil moisture response to rainfall (-5 to 5 weeks) and evapotranspiration (-10 to 5 weeks) anomalies in a matter of weeks are consistent with previous works showing soil moisture echoes rainfall anomalies in a range from days to weeks (Scaini et al., 2015; Martínez-Fernández et al., 2016), but also when driven by evapotranspiration (Otkin et al., 2013). Therefore, in a basin exposed to the high-energy characteristics of semi-arid climates, the weekly scale allows to diagnose disregarded short-term interactions, such as the ones of ETDI and SMDI on SPI, without losing resolution in the identification of mid to long-term interactions. The need to apply a weekly scale for the SPI index, which is barely used below the monthly scale, is recommended not only for the analysis of rainfall-evapotranspiration interactions but also for soil moisture ones which are often assessed over the monthly scale. Using the weekly scale for drought assessment demands an increase in spatial resolution. This is the reason why LSM using coarse inputs, such as the semi-aggregated physical characteristics of the basin, may generate results with scale effects. In fact, scale effects due to the coarse resolution of the spatial characteristics can increase the temporal scale at which the processes are effectively correlated (Rodriguez-Iturbe et al., 2001). Therefore, lag analysis needs to compile datasets of appropriate spatial

and temporal scales for their aims, like the high-resolution RS datasets evaluated in this study or LSM ones based on completely distributed data.

**6.2 Interpretation of the interactions between drought indices**

Adopting specific drought indices for rainfall, evapotranspiration and soil moisture allows exploring the interactions between variables of different levels of the land-atmosphere system. The pertinence of using the SPI, ETDI and SMDI to evaluate the

interactions between the variables' anomalies can be the subject of discussion depending on the specific advantages, drawbacks, and applicability of each index. There can be alternative drought indices even for soil moisture and evapotranspiration of better stability and statistical characteristics worth exploring. Furthermore, assessing the interactions of anomalies may be possible without using drought indices. However, since the commonplace in the analysis of the anomalies behind drought has been widely based on drought indices for comparable interpretation, we consider the SPI, ETDI and SMDI represent a set of comprehensive

indices to flexibly evaluate interactions at different time scales.

Fig. 8 a), and graphically 8 b), summarize the annual mode of interactions between the SPI, ETDI and SMDI. In the short to mid-term, both the ETDI and SMDI interactions with SPI concur on having moderate significance, with only a few negative and positive low interactions in the mid to long term. Positive correlations around lag 0 of the ETDI and SMDI with SPI indicate direct precedent

dependence of the indices, which means changes on ETDI and SMDI correlate positively (negatively) to positive (negative) changes on the SPI. The short-term correlations after rainfalls for both ETDI and SMDI (lagged response of these indices to SPI) are straightforward and were reported before in similar Iberian regions (Martínez-Fernandez et al., 2016). Sustained dry or wet anomalies in both variables favored by a positive correlation between indices are primarily restricted to a length of three seasons. Correlations beyond the year may represent the multi-annual persistence of anomalies common in Mediterranean climates.


The strong aggregation impact occurring when adopting the SPIw-26 and SPIw-52 indicates the analysis of interactions may be uncertain when indices are aggregated beyond the seasonal scale. It is evaluated if the magnifying effect of clusters with long aggregation periods (SPIw-26 and SPIw-52) is the autocorrelation of the indices. Significant autocorrelated values extend always for less than the period of aggregation of SPIw-i (2 weeks on SPIw-1, 3 weeks on SPIw-4, 10 weeks for SPIw-13, 18 on SPIw-26,

35 on SPIw-52, 40 on SPI-78, 45 on SPIw-104). Hence, we presume autocorrelation values of the SPI series are only partly caused by the period of aggregation. The partial autocorrelation of both ETDI and SMDI shows mostly two significant lags, which indicate the existence of an autoregressive model of type AR(2) (Fig. S3). An AR(2) may articulate the combination of growing and decaying factors behind the shifting balance of positive and negative correlations between indices. In this way, the clusters of interactions identified in Figs 3 to 6 a and b can be attributed to interactions between rainfall, evapotranspiration and soil moisture

anomalies. Conversely, clusters identified in series with a period of aggregation over the seasonal scale may be misleading for the analysis of drought interactions in Mediterranean environments. Thus, drought evolution in semi-arid environments can be considered as an expression of quick exchanges between land-atmosphere variables which develop within the seasonal scale. This implies the temporal resolution of drought indices must adopt short time scales (i.e., weekly scale) to capture the onset of drought.

The existence of the precedent influence of ETDI and SMDI on SPI (clusters of the negative range of lags) implies some unequal reciprocity (feedback) between evapotranspiration and soil moisture with rainfalls. This precedent influence is weaker than the influence of SPI on subsequent ETDI and SMDI anomalies (lagged response), but still remarkable. It is reasonable that the lagged response of evapotranspiration and soil moisture to rainfall is stronger and more long-lasting than the precedent influence (Fig. 8

a). Furthermore, the precedent influence period between the ETDI and SPI is stronger and of longer duration than the one of SMDI on SPI. This asymmetry suggests that ETDI, more than SMDI, has a weekly to seasonal precedent influence on rainfall (Fig. 8b). We expected a longer period of positive correlations of SMDI influencing rainfalls, given the multiple reports of soil moisture inducing memory to the near-surface atmosphere (Manning et al., 2018).

One reason why the ETDI shows a longer influence on SPI than the SMDI may be that ETDI from MOD16A2 is fed by the whole depth of soil moisture, while SMDI based on SMOS1km is limited to the top 5 centimeters of soil moisture, a very exposed soil level in semi-arid climates. The complexity of soil moisture dynamics, which barely follow a cyclic interaction (Rodriguez-Iturbe et al., 1991), can also explain a weaker relationship between the SMDI and SPI compared to the ETDI. Other reasons for this may be in the prevalence of maritime advection as the main contributor to evapotranspiration in the IP (Gimeno et al., 2010) compared to the prevalence of local soil moisture recycling common in more continental areas of Europe (Bisselink and Dolman, 2008). The advective explanation is supported by the contrast between the few weeks of precedent influence of soil moisture on rainfall we observe in the Ebro basin and the up to 250 days of precedent influence of continental areas prone to soil moisture recycling (Rowntree and Bolton, 1983; Bisselink and Dolman, 2008). Some studies focused on continental climates of relevant summer rainfall have described the implications of the alteration of the recycling due to soil moisture depletion during heatwaves and drought which can eventually alter the atmosphere (Rasmijn et al., 2018; Miralles et al., 2019). In the Mediterranean climate of the Iberian Peninsula characterized by the lack of summer rainfall, soil moisture annually reaches such low levels that we can expect annual summer alterations in the near atmosphere. Differences between areas where soil moisture plays a role, like central Europe, and areas where soil moisture is unable to control the evolution of the system under high-energy conditions, like the Iberian Peninsula, have been reported before in Mediterranean-like Western Australia (Herold et al., 2016).

In consequence, our results at the Ebro basin seem compatible with the frequent activation of a reinforcing or self-intensification loop (Brubaker and Entekhabi, 1996), by which the precedent influence of negative (eventually positive) anomalies of evapotranspiration reducing (increasing) rainfall cascades into a depletion (rise) of soil moisture that further limits (enhance) the response of evapotranspiration restarting the cycle (Fig. 8c, right column). The weak precedence of soil moisture on rainfall compared to that of evapotranspiration expresses the limited duration of the control capacity of the soil moisture over evapotranspiration in semi-arid climates of the Mediterranean type (left column of Fig. 8c). Negative correlations when indices differ in sign (r<0 conditions, Fig. 8a) can be indicative of transitional periods of mid-seasons. The sharp shift from the cluster of short-term positive correlations to the cluster of mid- to long-term negative correlations suggests a limit in the persistence of the self-intensification mechanism. A physical interpretation of the shift may be related to the change in the dominance of the sequence from the one under high-energy conditions (Fig. 8c, right column) to the one under low-energy conditions (Fig. 8c, left column). A low-energy inhibiting mechanism was already described by Brubaker and Entekhabi (1996) of negative correlations between soil moisture and surface temperature under low temperatures. Given the direct link between evapotranspiration and temperature, the shift from positive to negative interactions happening at time scales over the semester but below the year scale suggests that the arrival of winter low-energy conditions terminates the dominance of the self-intensifying loop of evapotranspiration.

In this way, the annual cycle can be modelled as the seasonal succession of two sequences: one under the low-energy conditions of winter when evapotranspiration no longer outweighs the inhibiting of soil moisture due to rainfall (left column of Fig. 8c), and the other under high-energy conditions driven by evapotranspiration (right column of Fig 8c). The shift between the long period of interactions dominated by evapotranspiration (right column of Fig. 8c) and a short period of interactions controlled by rainfall

and soil moisture (lower sequence of Fig. 8c) is generally driven by an energy threshold. However, certain levels of rainfall and soil moisture anomalies may temporally advance or delay the shift. This reason explains why under high-energy conditions drought may terminate due to heavy rainfall, while soil moisture deficits under low-energy conditions may cause an anticipated onset of the self-intensification loop of evapotranspiration.

The conceptualization of the interactions illustrated in Fig. 8 aims to raise awareness about the power of evapotranspiration anomalies to alter the land-atmosphere system, year-round, beyond hydro-meteorological extremes (Seneviratne et al., 2006; Otkin et al, 2013; Teuling, 2018; Miralles et al., 2019). Another reason supporting the year-round implications of the dominance of evapotranspiration over soil moisture is that rainfall mostly transfers to evapotranspiration in semi-arid climates (Rodriguez-Iturbe et al., 2001), where the often-underestimated interception (Savenije et al., 2004) further increases evaporation at the expense of soil moisture. Additionally, as the semi-arid Mediterranean climate likely presents thresholds of rainfall, evapotranspiration and soil moisture anomalies different from those triggering hydrometeorological extremes in other areas (Tramblay et al., 2021), the evapotranspiration-dominated sequence may initiate more often, but also more abruptly, than in regions of lower energy inputs. All these aspects, together with the increasing chance of extremes in the Mediterranean area due to climate change (Samaniego et al., 2018), recommend assessing changes in the balance of land-atmosphere interactions from the basis of this study.

However, we bear in mind that our results may oversimplify the causality, since processes not analyzed in this study may also play a role. The multiple periods showing neither prevalent positive nor negative correlations between indices indicate a loss of linear interaction. A source of non-linearity is vegetation due to its mediating role in water exchanges of the land-atmosphere system. Plants can control evapotranspiration and soil moisture in adaptation to water stress in non-linear manners that depend more on the type of vegetation (Katul et al., 2012), particularly within the Mediterranean floras (Boulet et al., 2020), than on the evapotranspiration or soil moisture status. Vegetation can also modulate the partitioning of energy governing evapotranspiration (Lansu et al., 2020) but similar ones are reported with soil moisture (Barbeta et al., 2015). In consequence, the quick response to drought of rainfed crops and sclerophyllous vegetation (Vicente-Serrano et al., 2019) may obscure the interpretation of the links between rainfall, evapotranspiration and soil moisture. This is of concern to the ETDI and SMDI results because interactions of vegetation integrate the status of the atmospheric and the land-surface variables (Peters et al., 1991). Nonetheless, additional factors of uncertainty may arise from teleconnections such as the well-known NAO or WeMO (Barnston and Livezey, 1987, Conte et al., 1989) or the oceanic ones like AMO (Kerr, 2000) altering the land-atmosphere system at large scales.

### 6.3 The value of remote sensing and land-surface model's estimates

The RS and LSM results of the lag analysis of the ETDI – SPI interactions show consistently comparable results in contrast to the remarkable disagreement between RS and LSM for the SMDI - SPI interaction. Results of SMDI obtained with the LSM show substantially lower correlations than the ones of RS, while also differing in the timing of the clusters of correlation. We expected the opposite, that the LSM, as being simpler than reality, has stronger SPI – ETDI - SMDI correlations than the RS dataset. We assume the implicit accumulation of uncertainties of modelling (Rodriguez-Iturbe et al., 1991), partly inherited from inputs but also from LSM structure, is the cause of the decrease in correlations. This is particularly true for soil moisture, a variable integrating exchanges between climate, soil and vegetation (Rodriguez-Iturbe et al., 2001). Secondly, this is an offline simulation, where the atmosphere (SAFRAN) is forcing the land-surface (SURFEX-ISBA) without explicit feedback of SURFEX-ISBA influencing back SAFRAN. SAFRAN estimates real conditions by ingesting observations, so the feedback is implicit in results, which may be

insufficient to represent reality. Thirdly, the model itself does not consider important processes like the interactive response of vegetation. ISBA has an interactive vegetation module (ISBA-A-gs), but Mediterranean vegetation can be particularly challenging for it. We expect to test the capabilities of interactive modelling vegetation in a follow-up study. Uncertainties of ISBA with vegetation have also roots in the use of ECOCLIMAP2 database, which shows inaccuracies of cover type and LAI. ECOCLIMAP assumes the maximum/minimum LAI occur in June/February in contrast with the early spring and autumn LAI maximums characteristic of the Mediterranean environment (Queguiner et al., 2011). All in all, the differences between LSM and RS datasets are already an important result to improve the LSM and useful insight into the use of offline LSM drought simulations.

Our results positively verify that RS represents an effective tool to overcome the problem of sparsely observed soil moisture or evapotranspiration, whose crucial role in drought evolution requires high-resolution data similarly to precipitation (AghaKouchak and Nakhjiri, 2012). Including high-res evapotranspiration products from MODIS (MOD16A2) and soil moisture from SMOS missions (SMOS1km) together with the distributed rainfall reanalysis data allows dedicated interpretation of the interactions between these two drought-relevant variables and rainfall, and their role in the water balance of the land-surface interface (Dai, 2011). Especially for evapotranspiration, the maps and series of LSM SURFEX-ISBA are comparable to those of RS, which supports the reliability of LSM despite their limited capability in arid regions (De Kauwe et al., 2015).

The temporal and spatial patterns of the anomalies are overly identified both by RS and the LSM model. The RS data seems able to capture a more complex scheme of interactions than the LSM model, despite the intrinsic data issues of the RS sensors and the performance of the algorithms used to generate the products. Conversely, the LSM seems sensitive to uncertainties from input data, especially surface properties, and the offline forcing. The parametrization of the model assumes a semi-distributed approach by sub-basins of the catchment on which each sub-basin is defined based on average values of land over and soil characteristics of the ECOCLIMAP database, which may induce some patchiness of LSM results compared to the RS results. The offline run means that the meteorological data forces the LSM, but the feedbacks are lost beyond the meteorological observations included as observation in the model. Additional aspects can be, for instance, the impact of groundwater redistributing soil moisture depending on topography, which is underrepresented in the LSM. Limitations of LSM have been reported in multiple works before and are subject of improvement (Teuling et al., 2006; Samaniego et al., 2018). Either way, uncertainties are causes of major concern in the lag analysis where they can alter the correlations between indices and obscure the interpretation of the interactions.

These aspects together can be critical to improving evapotranspiration and soil moisture estimates (Ukkola et al., 2016). Solving these inaccuracies would increase the value of RS and LSM estimates. This study exemplifies the potential of high-resolution RS and LSM products for a wide range of applications, such as drought analysis.

## 7 Conclusions and perspectives

The analysis of droughts in the Ebro basin using dedicated evapotranspiration and soil moisture drought indices based on high-resolution data from MOD16A2, SMOS1km and the LSM SURFEX-ISBA provides the following insights.

The monthly scale commonly adopted for drought evaluation (e.g., SPI-3) may overlook the quick evolution of drought from an agricultural and environmental perspective, especially in the high-energy climates of the Mediterranean basin where the anomalies of rainfall, evapotranspiration and soil moisture can vary in a matter of days. ETDI shows the strongest response at a weekly scale

while it remains also influential in the mid-term. SMDI can also quickly evolve with anomalies of evapotranspiration and particularly with lasting anomalies of rainfall. The weekly scale is advantageous to describe trends and shifts in the evolution of the indices and to identify disregarded interactions such as the preceding influence of ETDI on SPI.

The ETDI and SMDI indices, together with the SPI adapted to the weekly scale, allow tracking of the evolution of the anomalies of evapotranspiration, soil moisture and rainfall, as well as their interactions driving water anomalies in the region. There is great consistency between the time series of ETDI, SMDI and SPI. Lag analysis between these indices clarifies the interactions between anomalies on different levels of the surface-atmosphere system, information that is neglected when using multivariable indices or indices aggregated beyond the seasonal scale. The lag analysis also identifies sequences of interactions defining reinforcing or
inhibiting feedbacks. Evapotranspiration dominates the water balance of the Iberian semi-arid climate, especially during high-energy periods. This dominance frequently exceeds the controlling action of rainfall and soil moisture, inducing the reinforcing dry loop. Because of the relevance of evapotranspiration, heat waves further fueling dry events deserve further attention. The weak influence of soil moisture on subsequent evapotranspiration and rainfall limits its capability to control the propagation of anomalies.

RS datasets of MOD16A2 and SMOS-1km accurately estimate the temporal and spatial anomalies in the basin. Evapotranspiration from the LSM SURFEX-ISBA closely resembles the RS one of MOD16A2. Results differ substantially between SMOS1km and SURFEX-ISBA estimates of soil moisture. RS uncertainties arise mainly from data gaps. Land-surface model's estimates can extend the evaluation of soil moisture beyond the surface towards the root zone but face notable challenges from offline simulation neglecting feedback, as well as from the quality of input data that defines surface characteristics. RS outcompetes the LSM in the
ability to integrate information about challenging processes, such as vegetation dynamics. Assimilation seems the way forward to integrate the best aspects of both kinds of data. For as long as ground-based observations remain sparse, RS and LSM represent effective tools to assess the water anomalies of the land-atmosphere system and their interaction mechanisms.

**Acknowledgements**

Funding: This work was performed at Hydrology and Climate Change lab of Pere Quintana-Seguí at Ebro Observatory within the
Project HUMID project [CGL2017-85687-R] which was supported by the Spanish Agency of Research (AEI), in the framework of AEI/FEDER-EU grants.

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

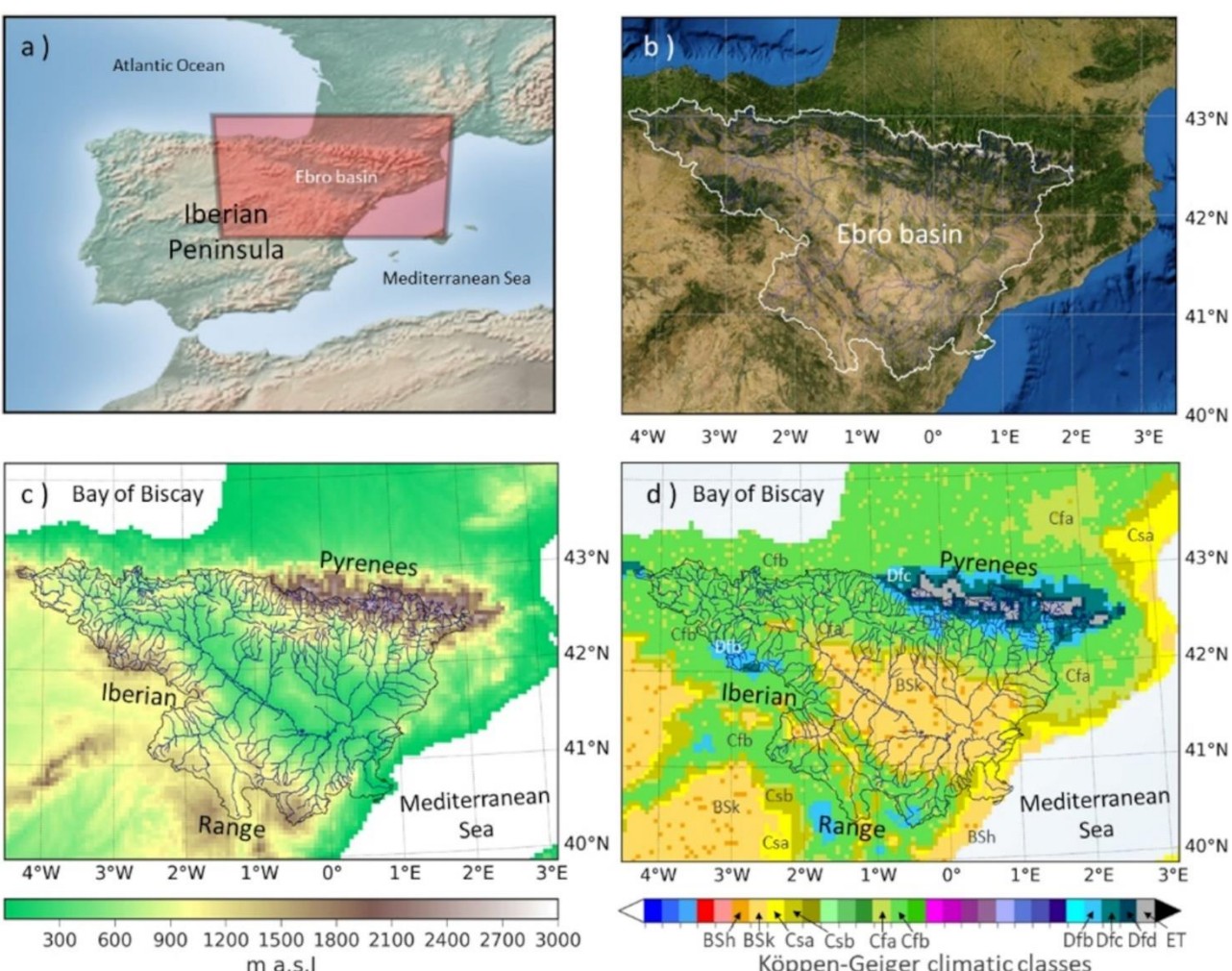

**Figure 1: a) Ebro basin location in the Iberian Peninsula (OSM OpenTopoMap). b) Land cover of the basin describes a contrasted basin between the forested areas of the mountains and the steppes of the central depression (© ESRI satellite online maps, World Imagery, 2022). c) Altitudinal range of the basin (IGN ES/FR MDT25, CC-BY 4.0). D) Climatic classes present in the basin according to the climatic classification of Köppen-Geiger (from data of Beck et al. (2018)).**

**2010-2017 - Monthly scale**

|  |  | ETDI | | SMDI | |
|---|---|---|---|---|---|
|  |  | RS | LSM | RS | LSM |
| ETDI | RS |  | 0,77 | 0,58 |  |
|  | LSM | 0,77 |  |  | 0,32 |
| SMDI | RS | 0,58 |  |  | 0,27 |
|  | LSM |  | 0,32 | 0,27 |  |
| SPI | SPIm-1 | 0,39 | 0,51 | 0,42 | 0,45 |
|  | SPIm-3 | 0,62 | 0,80 | 0,61 | 0,41 |
|  | SPIm-6 | 0,72 | 0,78 | 0,58 | 0,30 |
|  | SPIm-12 | 0,81 | 0,71 | 0,63 | 0,24 |

**2010-2017 - Monthly scale - Dry periods**

|  |  | ETDI | | SMDI | |
|---|---|---|---|---|---|
|  |  | RS | LSM | RS | LSM |
| ETDI | RS |  | 0,60 | 0,50 |  |
|  | LSM | 0,60 |  |  | 0,43 |
| SMDI | RS | 0,50 |  |  | 0,13 |
|  | LSM |  | 0,43 | 0,13 |  |
| SPI | SPIm-1 | 0,23 | 0,37 | 0,52 | 0,59 |
|  | SPIm-3 | 0,42 | 0,67 | 0,41 | 0,32 |
|  | SPIm-6 | 0,45 | 0,39 | 0,19 | 0,17 |
|  | SPIm-12 | 0,75 | 0,53 | 0,40 | 0,22 |

**2010-2017 - Weekly scale**

|  |  | ETDI | | SMDI | |
|---|---|---|---|---|---|
|  |  | RS | LSM | RS | LSM |
| ETDI | RS |  | 0,55 | 0,50 |  |
|  | LSM | 0,55 |  |  | 0,21 |
| SMDI | RS | 0,50 |  |  | 0,19 |
|  | LSM |  | 0,21 | 0,19 |  |
| SPI | SPIw-4 | 0,57 | 0,68 | 0,57 | 0,33 |
|  | SPIw-13 | 0,63 | 0,74 | 0,51 | 0,21 |
|  | SPIw-26 | 0,57 | 0,60 | 0,40 | 0,16 |
|  | SPIw-52 | 0,48 | 0,45 | 0,43 | 0,15 |

**2010-2017 - Weekly scale - Dry periods**

|  |  | ETDI | | SMDI | |
|---|---|---|---|---|---|
|  |  | RS | LSM | RS | LSM |
| ETDI | RS |  | 0,34 | 0,46 |  |
|  | LSM | 0,34 |  |  | 0,18 |
| SMDI | RS | 0,46 |  |  | 0,12 |
|  | LSM |  | 0,18 | 0,12 |  |
| SPI | SPIw-4 | 0,30 | 0,49 | 0,57 | 0,25 |
|  | SPIw-13 | 0,32 | 0,47 | 0,36 | 0,29 |
|  | SPIw-26 | 0,21 | 0,30 | 0,40 | 0,02 |
|  | SPIw-52 | 0,31 | 0,16 | 0,39 | -0,03 |

\* Dry periods considered when SPI<0, ETDI<0, SMDI<0

**Table 2: Matrices of significant (in bold) correlation coefficients considering p-values=0.05 for pairs of indices at monthly and weekly scale for the period 2010-2017 of the SPI I, ETDI and SMDI series, and the corresponding dry period subsets. Higher correlation values show a more intense red color. Since the correlations of interest refer to comparing the same type of data sources for different indices, dark grey cells identify unsuitable combinations for the analysis such as comparing RS results from one index with LSM results from another index.**

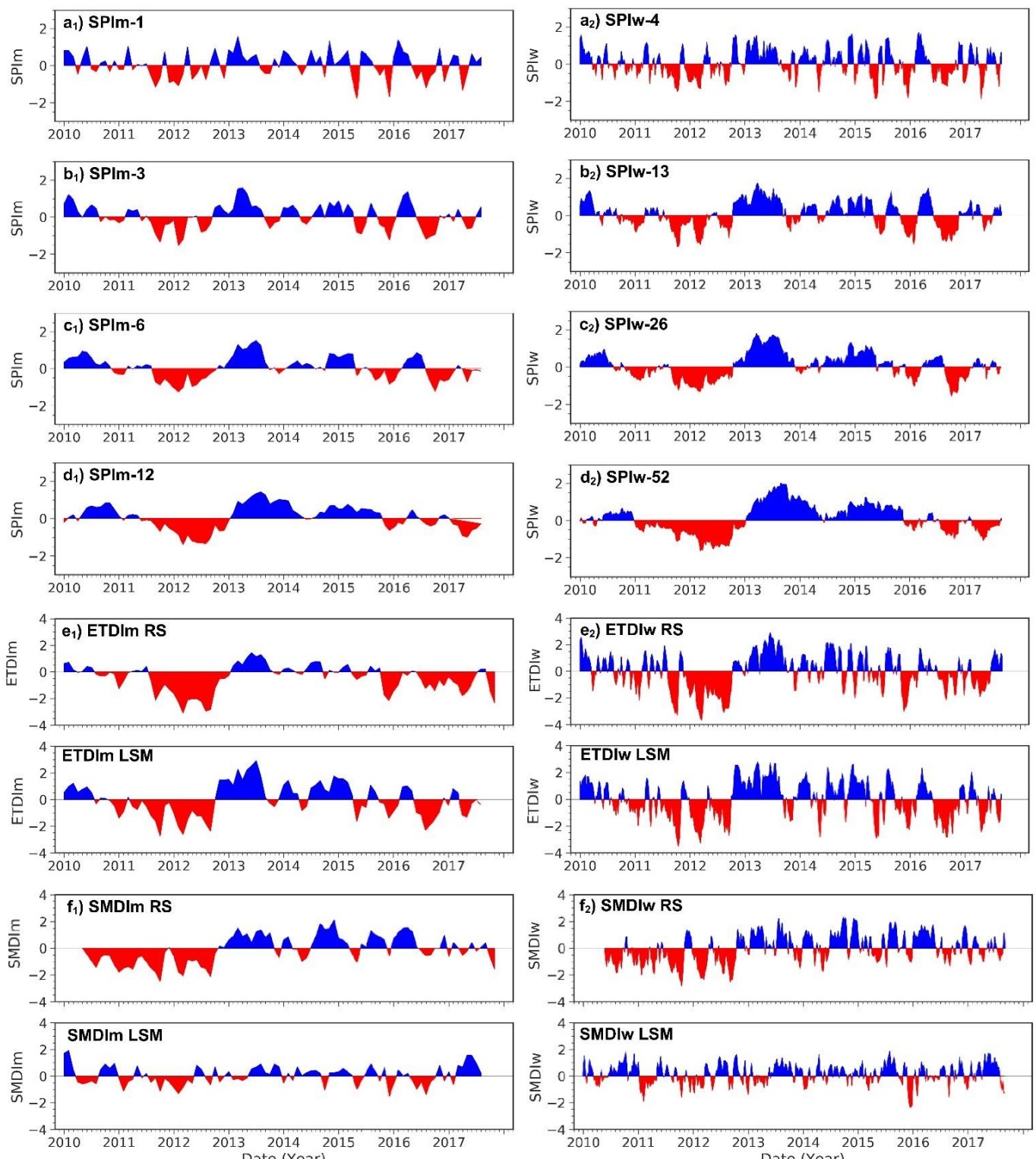

**Figure 2: Time series of SPI, ETDI and SMDI indices at monthly scale (left column of subplots identified with sub-index 1) and weekly scale (right column of subplots identified with sub-index 2). a) to d) correspond to SPIm-1, SPIm-3, SPIm-6 and SPIm-12, e) and f) display the temporal evolution of the ETDI and SMDI. These ones show two rows corresponding to the series based on RS and LSM data. Red /blue color represents dry /wet periods. Periods of drought occur when SPI<0, ETDI>0, SMDI<0.**

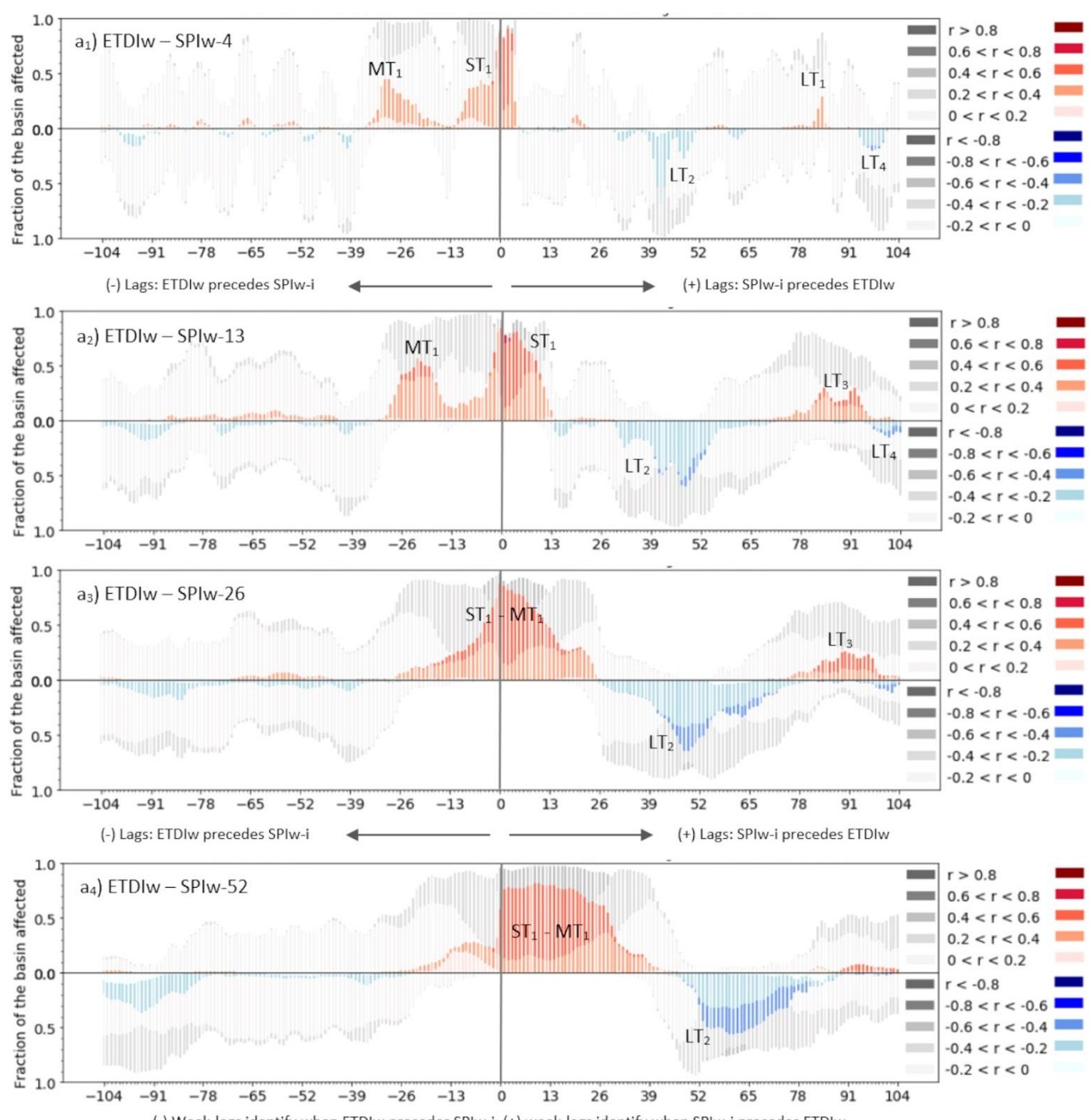

**Figure 3: Lag plots of SPI-1, SPI-3, SPI-6 and SPI-12 (expressed as SPIw-4, 13, 26, and 52 weeks respectively) with remote sensing (RS)**
**ETDIw index at weekly scale for the period 2010-2017. Lags are calculated for the -104 leading and +104 lagged time steps of ETDIw in reference to the SPIw-i. The height of the bars of the plot indicates the area of the basin affected by non-significant (grey scale) or significant (coloured scale) correlations. The saturation of the colored scale indicates the magnitude of the significant positive (red) or negative (blue) Pearson correlation coefficient of SPIw-i and ETDIw.**

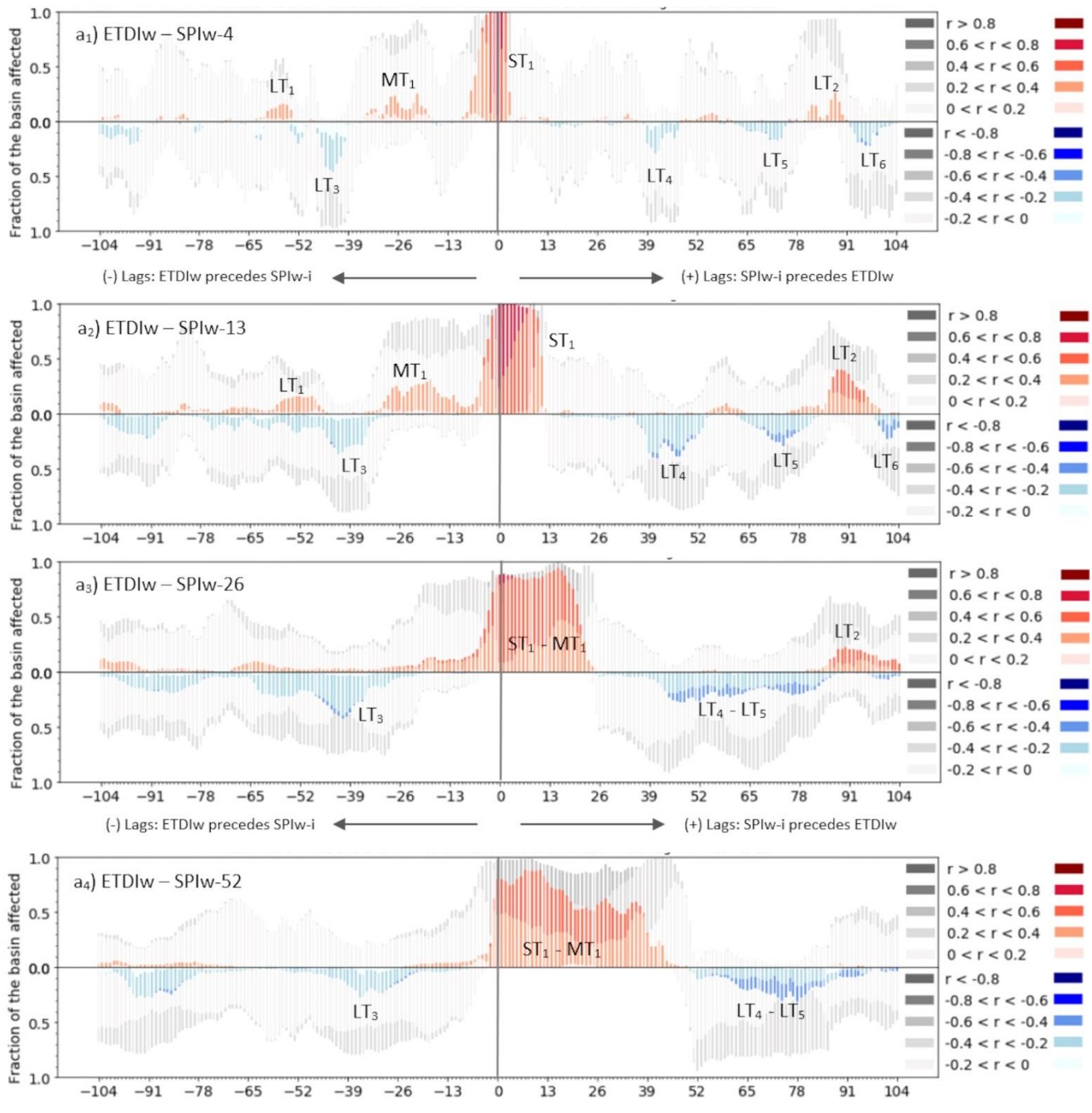

**Figure 4: Lag plots of SPI-1, SPI-3, SPI-6 and SPI-12 (expressed as SPIw-4, 13, 26, and 52 weeks respectively) with land-surface model (LSM) ETDIw index at weekly scale for the period 2010-2017. Lags are calculated for the -104 leading and +104 lagged time steps of ETDIw in reference to the SPIw-i. The height of the bars of the plot indicates the area of the basin affected by non-significant (greyscale) or significant (coloured scale) correlations. The saturation of the colored scale indicates the magnitude of the significant positive (red) or negative (blue) Pearson correlation coefficient of SPIw-i and ETDIw.**

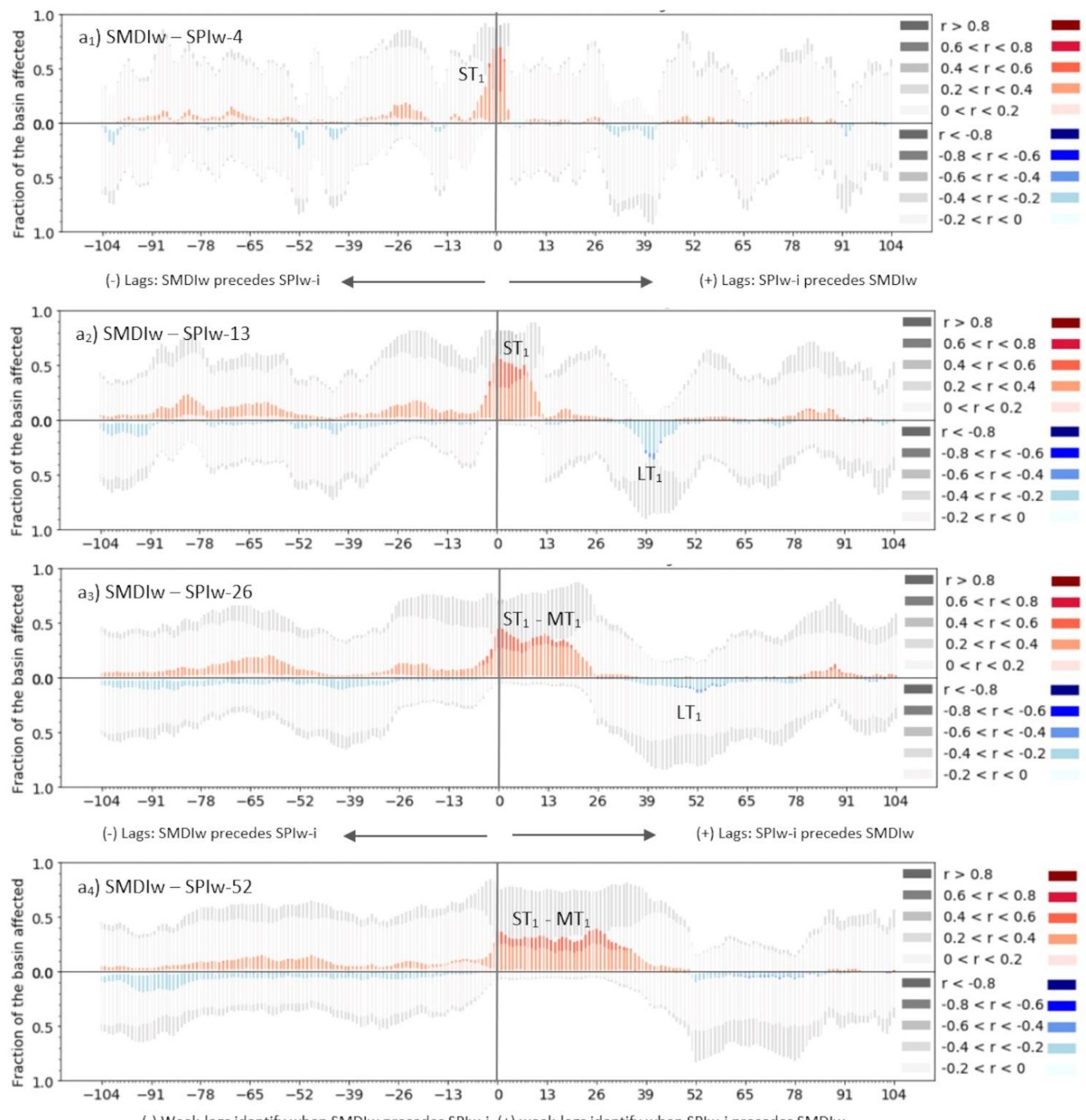

**Figure 5: Lag plots of SPI-1, SPI-3, SPI-6 and SPI-12 (expressed as SPIw-4, 13, 26, and 52 weeks respectively) with remote sensing (RS) SMDIw index at weekly scale for the period 2010-2017. Lags are calculated for the -104 leading and +104 lagged time steps of SMDIw in reference to the SPIw-i. The height of the bars of the plot indicates the area of the basin affected by non-significant (greyscale) or significant (colored scale) correlations. The saturation of the colored scale indicates the magnitude of the significant positive (red) or negative (blue) Pearson correlation coefficient of SPIw-i and SMDIw.**

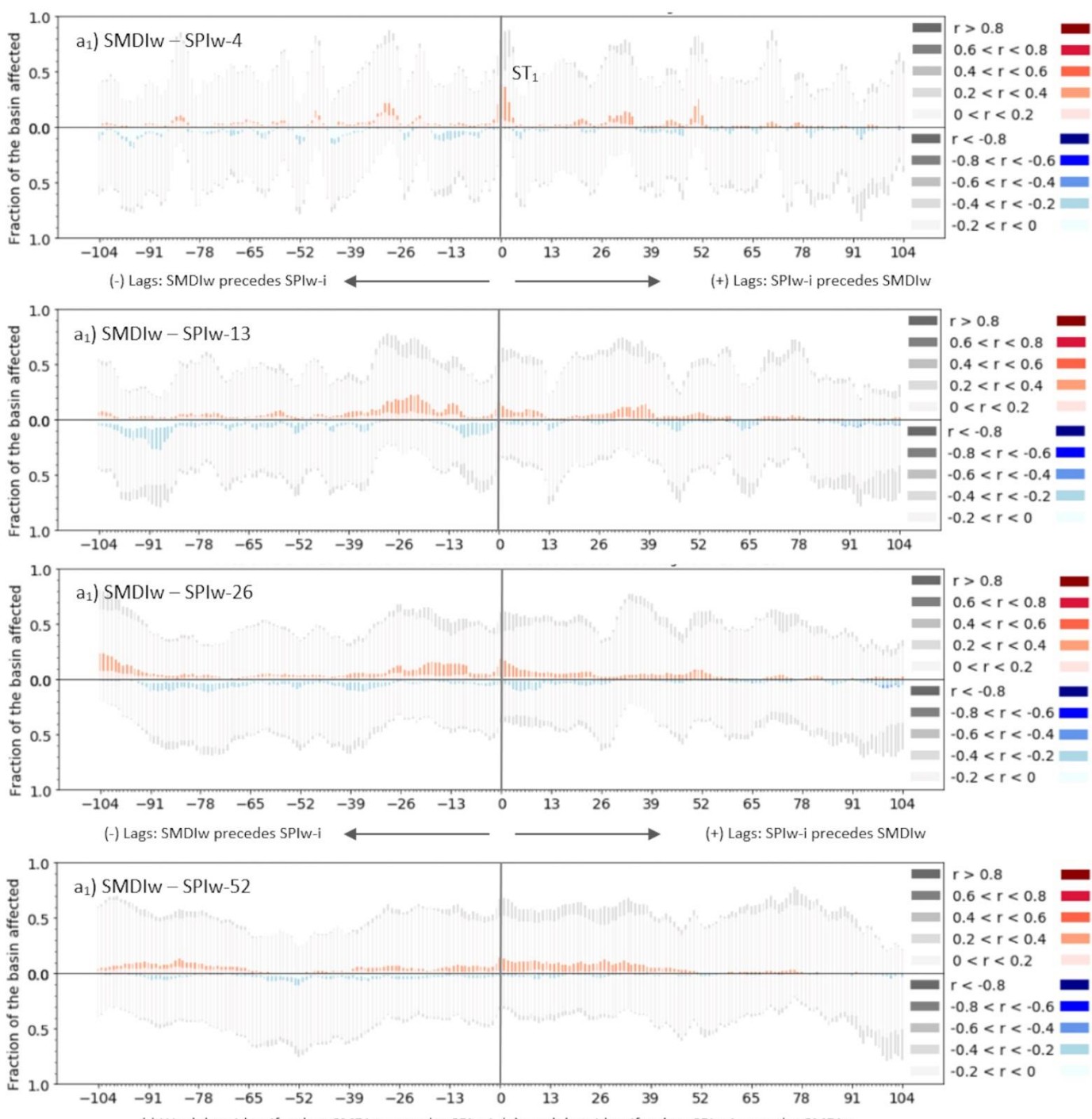

 **Figure 6: Lag plots of SPI-1, SPI-3, SPI-6 and SPI-12 (expressed as SPIw-4, 13, 26, and 52 weeks respectively) with land-surface model (LSM) SMDIw index at the weekly scale for the period 2010-2017. Lags are calculated for the -104 leading and +104 lagged time steps of SMDIw in reference to the SPIw-i. The height of the bars of the plot indicates the area of the basin affected by non-significant (greyscale) or significant (colored scale) correlations. The saturation of the colored scale indicates the magnitude of the significant positive (red) or negative (blue) Pearson correlation coefficient of SPIw-i and SMDIw.**

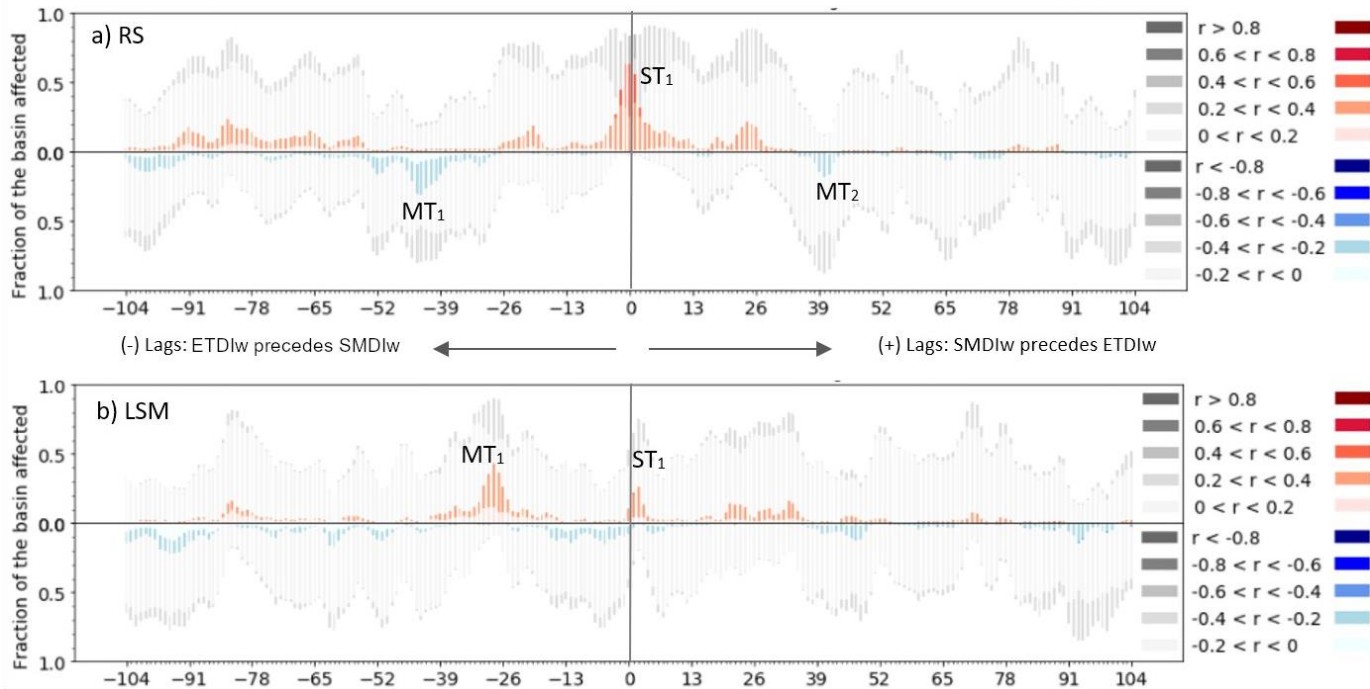

**Figure 7: Lag plots of ETDIw – SMDIw at weekly time scale in the period 2010-2017 for a) RS data and b) LSM data. Lags are calculated for the -104 leading and for the +104 lagged time steps of SMDIw in reference to theETDIw. The five levels of red and blue indicate the positive (red) or negative (blue) Pearson correlation coefficient between the ETDIw and the SMDIw for each time step (lead or lag time step). The height of the bars of the plot indicates the area of the basin affected by that level of r Pearson values.**

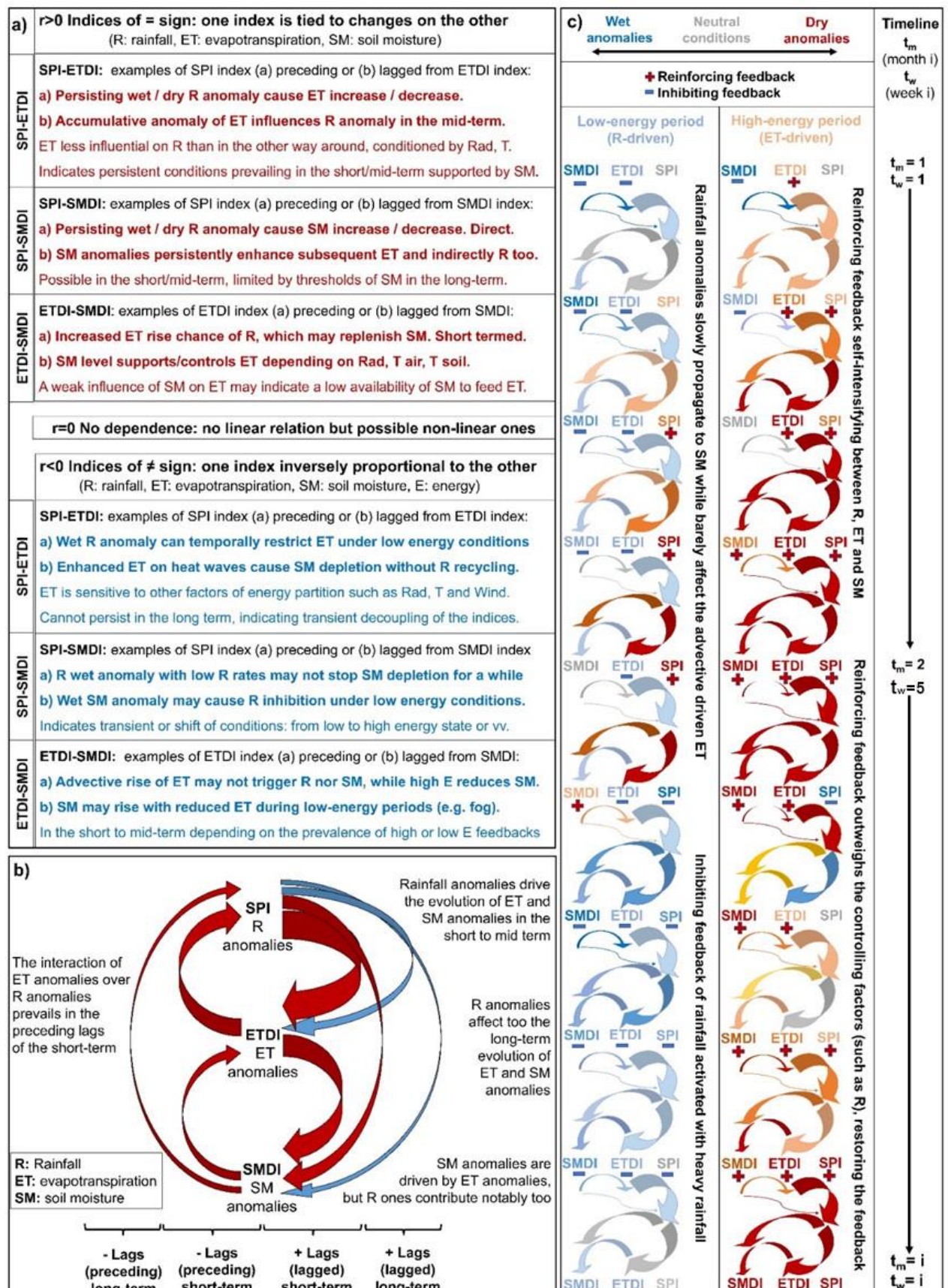

**Figure 8: (a) Interpretation of the correlations between SMDI-SPI, ETDI-SPI and SMDI-ETDI indices. The r>0 box exemplifies positive correlations between the indices (positive correlations of red arrows in b)) while the r<0 box defines the negative ones (negative**

970

correlations of blue arrows in b)). **(b)** Summary of the annual magnitude and timing of the interactions. Red arrows represent positive correlations while blue arrows the negative ones. Arrow width represents the magnitude of the correlation. Arrow direction determines the direction of the interaction. The timeline at the bottom indicates the scale of the interactions ranging from short-term to long-term. **(c)** Sequences of prevalent reinforcing (upper) and inhibiting (lower) conditions alternating during the annual cycle based on the scheme of interactions described in a) and b). The upper sequence is the self-intensifying loop driven by evapotranspiration under high-energy conditions. The lower sequence displays the inhibiting role of rainfall/soil moisture under low-energy conditions.

**Supporting material:**

|  | | Spatial dimension | | |
|---|---|---|---|---|
| r Pearson correlation values of full length and full area series vs. | | 10129px (full area) | 2580px (1/4 area) | 516px (1/20 area) |
| Temporal dimension | (1/2 length series) | 0.97 | 0.97 | 0.97 |
| | (1/4 length series) | 0.92 | 0.81 | 0.49 |
| | (1/8 length series) | 0.78 | 0.73 | 0.19 |

**Table S1: Impact of temporal and spatial subsets of the evapotranspiration dataset on the consistency of the ETDI series.**

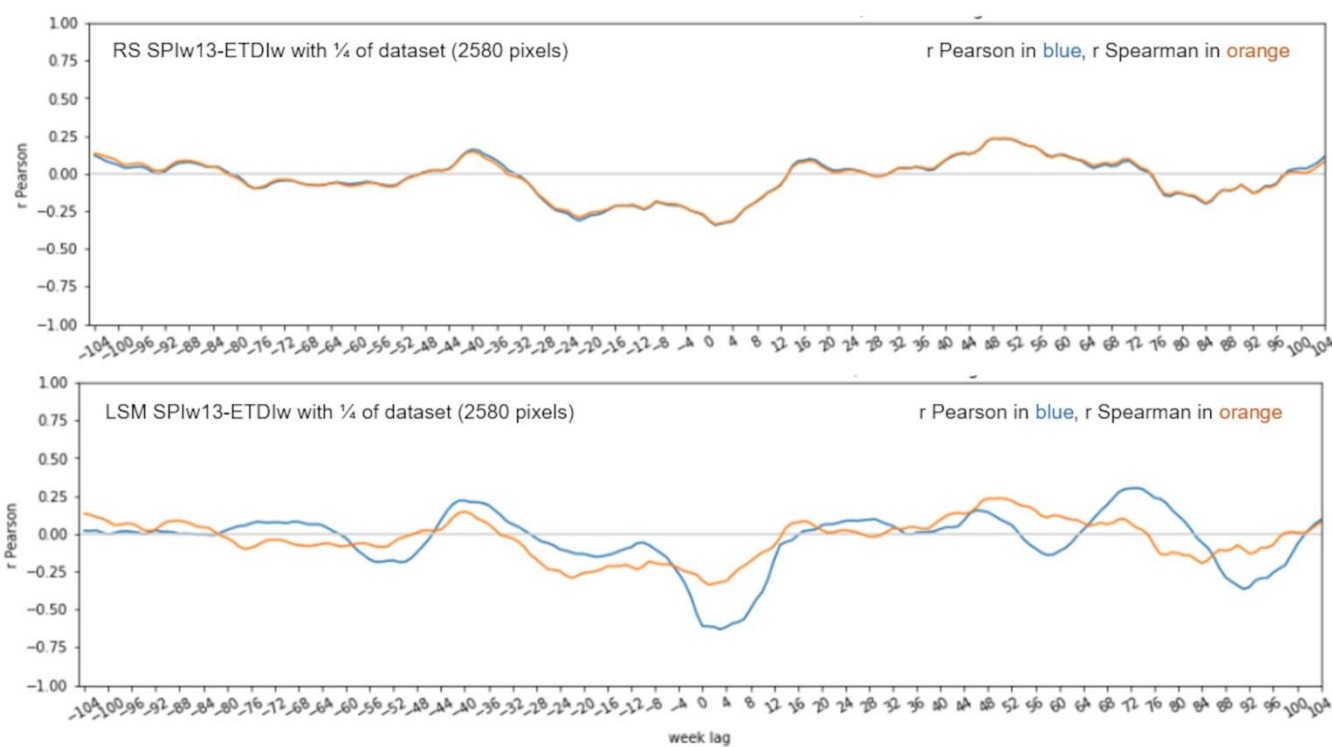

**Figure S2: Similarity of series of lags calculated with the r Pearson coefficient and r Spearman coefficient between SPIw13-ETDIw indices considering ¼ of the total dataset (2580 pixels) using (a) RS data and (b) LSM data. r Pearson series colored in blue, r Spearman in orange. Y axis identifies either r Pearson and r Spearman values of the correlation between indices.**

ETDIw

SMDIw

SPIw-1

SPIw-26

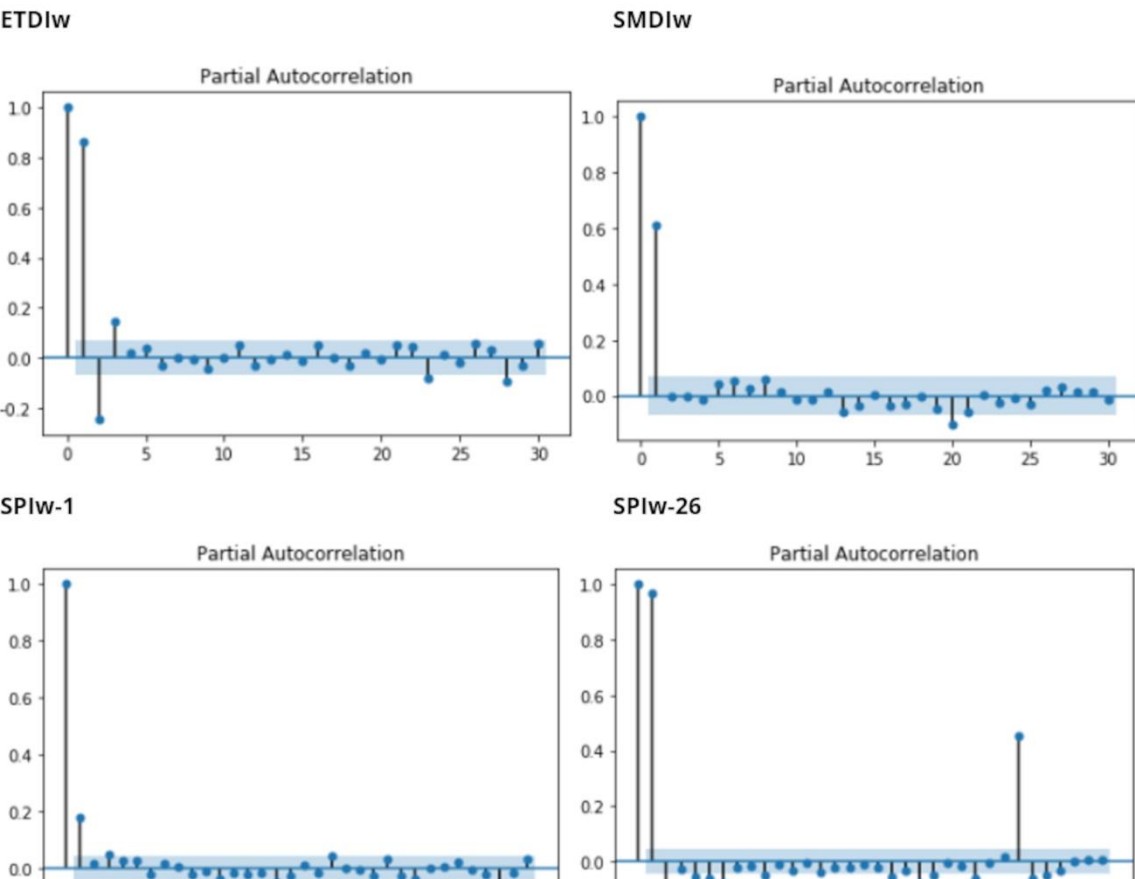

Figure S3: Partial autocorrelation results of the RS ETDI and SMDI series indicating a primary balance between the first two significant factors of autocorrelation. The interpretation of results suggests this partial autocorrelation plots depict the waving interplay between positive and negative correlations of each variable with other factors, which supports the definition of drought anomalies as an ever-changing balance of influence between the key variables of the land-atmosphere system.