# Peer review of "Interactions between precipitation, evapotranspiration and soil moisture-based indices to characterize drought with high-resolution remote sensing and land-surface model data."

_Natural Hazards and Earth System Sciences, 2022_

## Referee Comment (RC1)

REVIEW OF MANUSCRIPT NHESS-2022-65

The study addresses a drought analysis in the Ebro River basin in Spain by using remote sensing (RS) data of evapotranspiration from MOD16A2ET and soil moisture data from SMOS1km 20 as well as SURFEX-ISBA land-surface model (LSM) data to calculate the Evapotranspiration Deficit Index (ETDI) and the Soil Moisture Deficit Index (SMDI) for the period 2010-2017. Also, SAFRAN data are used to calculate the Standardized Precipitation Index (SPI) at different aggregation time scales. The drought indices are computed both at the monthly and weekly scales. In particular, the study investigates the mechanisms of drought propagation in the land-atmosphere system by analyzing the temporal lags between the drought indices to identify the synchronicity and memory of the anomalies between precipitation, evapotranspiration and soil moisture to interpret factors involved in drought onset.

Overall, the study is interesting and well written. A few comments follow.

1) In the Introduction, the bibliographic review of previous studies on the use of remote sensing products in drought analysis at the weekly scale focused on rainfed agriculture could be extended. I suggest referring to the following study and reference therein:

   https://doi.org/10.5194/nhess-20-471-2020

2) The complex interactions between drought indices, investigated by means of a throughout correlation analysis, highlights feedbacks among the considered variables, with a preeminent role of evapotranspiration in the link between rainfall and soil moisture. The study is carried out by calculating the Pearson correlation coefficients between pair of series of the three drought indices at weekly scale, introducing lags from -104 weeks to +104 weeks. The use of the Pearson correlation coefficient implies that the underlying variables are normal distributed. This is true, by definition, for the SPI, but what about the ETDI and SMDI? ETDI and SMDI series must be first checked for normality and, in case the normal hypothesis is rejected, the Spearman rank correlation statistic must be used instead of the Pearson correlation coefficient.

3) Lag analysis show a remarkable disagreement between RS and LSM for the SMDI - SPI interactions, with SMDI obtained with the LSM showing substantially lower correlations than the ones of RS, while also differing in the timing of the clusters of correlation. The authors state that they expected that the LSM, as being simpler than reality, had stronger SPI – ETDI - SMDI correlations than the RS dataset and justify this result by the accumulation of uncertainties of modelling, inputs and LSM structure. I suggest the authors to better argue this point. Why should the adopted LSM work only for SPI and ETDI? Furthermore, gridded soil moisture datasets are available at the global and European scale (see for instance the Copernicus climate data service). These datasets can be a valid alternative to the LSM soil moisture data and deserve some references in the study.

4) With reference to the feedback mechanisms depicted in Figure 8, the Discussion can be enriched by a comparison with previous studies investigating the same mechanism, such as:

   https://doi.org/10.1038/s41558-018-0154-5

   https://doi.org/10.1038/s41558-018-0138-5

   https://doi.org/10.1038/s41558-018-0114-0

5) Please check Figure 7 since you wrote the same thing for both negative and positive lags.

---

## Author Comment (AC1)

**Response to reviewer RC2 (**reviewer in black Open Sans 11, **response in blue, Calibri,11)**

" by Gaona et al. submitted to NHESS-discussion an analysis of the atmosphere-soil-vegetation interaction, performed through a time correlation analysis among indices of precipitation anomalies (SPI computed at weekly scale), evapotranspiration deficit index (ETDI) and soil moisture deficit index (SMDI) for the period 2010-2017. ETDI e SMDI input data are provided from both remote sensing and from modelling approaches. The study area is the Ebro basin (Spain). The goal of the work is to get more insights into the drought propagation mechanisms.

The manuscript is within the scope of the Journal and potentially of interest for the readers of NHESS. However, I have some main concerns that prevent from publishing the manuscript in its present form. Here below my general comments:

1.  I found very interesting the adopted methodology. My main concern is on the use in the specific case study of standardized indexes. The time span analysed is 8 years. This means that whatever the adopted method for standardization, the statistical population is 8 (maximum). In the original work by Narasimhan and Srinivasan (2005) the ETDI and SMDI are computed on a dataset covering a time span of 70 years (1911-1980), making robust the statistical approach necessary to compute SPIn (fitting of the gamma or Pearson III distribution), ETDI and SMDI (setting the range of variation through the definition of the min and max values, as well as the median to compute the deviation). In my opinion the authors should wide the database extending the time span to 2021 in order to perform an uncertainty analysis on the robustness of the adopted statistical approach.

    For example, it would be interesting to study the variability of the fitting for SPI and of the min-med-max values necessary to compute the ETDI and SMDI by considering n subset of n-1 elements (12 subset of 11 y data if you consider the time span 2010-2021) and studying how the statistical metrics and the indexes themselves vary in relation to the subset. I know that it is a lot of work, but in my opinion, this is mandatory to ensure a sound and robust time lag analysis. Therefore, my concerns are not on the methodology adopted for the analysis of the relationships among the indexes, but on the indexes themselves."

We sincerely appreciate the rigorous and accurate comments of the reviewer about the temporal constraints of the study and the sub-optimal application of drought indices for short time series. We are aware of the narrow timespan available for the analysis of soil moisture data but SMOS1km is the longest reliable data series available. Certainly, there are remote sensing databases with long-term time series of the variables of this study, even for soil moisture (e.g. CCI). However, apart from the coarse spatial resolution of CCI (0.25º instead of the 1km of SMOS1km), CCI inherits multiple inhomogeneities due to the different remote sensing data sources used to generate such long series. ASCAT and AMRS-E have been tested in the area of study in comparison with SMOS1km and despite their notable performance, SMOS1km outperforms them both in resolution and due to its lack of roughness and vegetation effects (Escorihuela and Quintana-Seguí, 2016). The SMOS1km dataset generated using the DisPATCh Algorithm (Merlin et al., 2013) has been also reported of remarkable reliability in similar studies in the area (Dari et al., 2021). Equal resolution SMAP and Sentinel-1 options are of much shorter data series and consequently not appropriate for the focus on lags of this study. Therefore, SMOS1km

dataset was the longest and optimal remote sensing option for soil moisture analysis. In the case of evapotranspiration remote sensing data, MOD16A2 is also the best option considering the spatial or the temporal constraints of the alternative databases of longer duration or similar spatial resolution (e.g. ERA5Land of 0.1x0.1º resolution spatial resolution, GLEAM 0.25x0.25º, Tomas-Burguera et al., 2019 only spans to 2014…etc.). Provided these constraints, we support the validity of considering SMOS1km + MOD16A2 as well as ETDI and SMDI as the best available options for the analysis of interactions of the selected variables, to evaluate the interactions between rainfall, evapotranspiration, and soil moisture.

Regarding indices, and following the previous clarification, the study aims to underline the convenience of addressing single-variable analysis of drought factors to promote the understanding of water exchanges under drought instead of discussing global indices only able to partially characterize drought but not specific mechanisms of interaction. That's the reason for adopting individual indices to assess the anomalies of relevant variables involved in drought's evolution. In view of this, there are multiple aspects of incompatibility between drought indices focused on different variables. In fact, the SMDI and ETDI are among the few indices defined for different relevant variables of the system exactly in the same way, which would be an advisable aspect for studies focused on the multivariate analysis of drought.

Following the recommendation of the reviewer, we tested the sensitivity of the ETDI and SMDI (they have the same definition based on calculating max, min, median annual weekly values of the data) to the number of years of data used to obtain the maximum, minimum and median values of the series necessary to calculate the indices. We adopted the ETDI because it has a longer series (~18years) than SMDI (~8 years) so that the results of half the length of the series of ETDI can give an idea of how the restricting length of the SMDI series impacts the outcomes of the study. Firstly, taking the full spatial dataset (all pixels of data: 10129px) we constrained the temporal data available to generate the mean annual max/med/min weekly values of the series to a half, a quarter and an eighth of the length of the series. The comparison of the correlation between the ETDI obtained using the full spatial and temporal dimension compared to the half, quarter and an eight length of the series using ¼ of the data (case A) indicate there is little impact due to the shortening of the timeseries. Adopting the length of 8 years of SMDI compared to 18 years would keep ETDI and SMDI values above r=0.95. Even an eighth of the 18y series length keeps r=0.78. The reason for such low impact is the high spatial resolution of the dataset, whose spatial heterogeneity contributes to compensate the risk of sensitivity of the ETDI/SMDI series to the shortening of the temporal dimension. This statement is supported by the results of the increased sensitivity of the ETDI to the shortening of the series when a fraction of the pixels instead of the full range of basin pixels is used (case B, with 1/20 of the pixels). Using 1/4 of all pixels (case B, not shown) causes a slight degradation of the series, while using a 1/20 fraction (case C) causes a sharp decline in the correlation, particularly for the biggest shortening of the timeseries (1/8 of length). Therefore, we can assume the ETDI and SMDI series are sufficiently representative despite their short length of the dataset thanks to the spatial resolution providing enough heterogeneity of the MOD16 and SMOS values to define sufficiently representative max, median and min values to generate the ETDI and SMDI series.

| SENSITIVITY ANALYSIS OF ETDI | | Spatial dimension | | |
|---|---|---|---|---|
| r Spearman coefficient | r Pearson of full time series vs. | 10129px (full) | 2580px (1/4) | 516px (1/20) |
| **Temporal dimension** 1/2 length | | 0,97 | 0,97 | 0,97 |
| 1/4 length | | 0,92 | 0,81 | 0,49 |
| 1/8 length | | 0,78 | 0,73 | 0,19 |

**A)   ETDI sensitivity to series length with full spatial dataset (10129 px)**

[Figure]

**C)   ETDI sensitivity to series length with 1/20 of the spatial dataset (516 px)**

[Figure]

2. "In my opinion, results on table 1 could be presented in a more effective way. I suggest presenting four different correlation matrixes (2010-2017m, dry periods m, 2010-2017 w, dry periods w). Each matrix has on the rows [ETDI RS; SMDI RS; SPIm-1; SPIm-3; SPIm-6; SPIm-12] and on the columns [ETDI RS; ETDI LSM; SMDI RS; SMDI LSM]. A colour code to highlight the Pearson correlation, ranging [0,1] would help the readability of the tables, supporting the presentation of the outcomes."

We thank the reviewer for the suggestions to improve the visualization of Table 1 which has been completely reformed following the indications. We realize the matrix format and the color code helps interpreting the magnitude of the correlation. (At Lines 845-850).

**2010-2017 - Monthly scale**

| | ETDI RS | ETDI LSM | SMDI RS | SMDI LSM |
|---|---|---|---|---|
| ETDI RS | | 0,77 | 0,58 | |
| ETDI LSM | 0,77 | | | 0,32 |
| SMDI RS | 0,58 | | | 0,27 |
| SMDI LSM | | 0,32 | 0,27 | |
| SPIm-1 | 0,39 | 0,51 | 0,42 | 0,45 |
| SPIm-3 | 0,62 | 0,80 | 0,61 | 0,41 |
| SPIm-6 | 0,72 | 0,78 | 0,58 | 0,30 |
| SPIm-12 | 0,81 | 0,71 | 0,63 | 0,24 |

**2010-2017 - Monthly scale - Dry periods**

| | ETDI RS | ETDI LSM | SMDI RS | SMDI LSM |
|---|---|---|---|---|
| ETDI RS | | 0,60 | 0,50 | |
| ETDI LSM | 0,60 | | | 0,43 |
| SMDI RS | 0,50 | | | 0,13 |
| SMDI LSM | | 0,43 | 0,13 | |
| SPIm-1 | 0,23 | 0,37 | 0,52 | 0,59 |
| SPIm-3 | 0,42 | 0,67 | 0,41 | 0,32 |
| SPIm-6 | 0,45 | 0,39 | 0,19 | 0,17 |
| SPIm-12 | 0,75 | 0,53 | 0,40 | 0,22 |

**2010-2017 - Weekly scale**

| | ETDI RS | ETDI LSM | SMDI RS | SMDI LSM |
|---|---|---|---|---|
| ETDI RS | | 0,55 | 0,50 | |
| ETDI LSM | 0,55 | | | 0,21 |
| SMDI RS | 0,50 | | | 0,19 |
| SMDI LSM | | 0,21 | 0,19 | |
| SPIw-4 | 0,57 | 0,68 | 0,57 | 0,33 |
| SPIw-13 | 0,63 | 0,74 | 0,51 | 0,21 |
| SPIw-26 | 0,57 | 0,60 | 0,40 | 0,16 |
| SPIw-52 | 0,48 | 0,45 | 0,43 | 0,15 |

**2010-2017 - Weekly scale - Dry periods**

| | ETDI RS | ETDI LSM | SMDI RS | SMDI LSM |
|---|---|---|---|---|
| ETDI RS | | 0,34 | 0,46 | |
| ETDI LSM | 0,34 | | | 0,18 |
| SMDI RS | 0,46 | | | 0,12 |
| SMDI LSM | | 0,18 | 0,12 | |
| SPIw-4 | 0,30 | 0,49 | 0,57 | 0,25 |
| SPIw-13 | 0,32 | 0,47 | 0,36 | 0,29 |
| SPIw-26 | 0,21 | 0,30 | 0,40 | 0,02 |
| SPIw-52 | 0,31 | 0,16 | 0,39 | -0,03 |

* Dry periods considered when SPI<0, ETDI<0, SMDI<0

Table 1: Matrices of significant (in bold) correlation coefficients considering p-values=0.05 for pairs of indices at monthly and weekly scale for the period 2010-2017 of the SPI I, ETDI and SMDI series, and the corresponding dry period subsets. Higher correlation values show a more intense red color. Since the correlations of interest refer to comparing the same type of data sources for different indices, dark grey cells identify unsuitable combinations for the analysis such as comparing RS results from one index with LSM results from another index.

3. "Figures 4-7. These are the core business of the work, but the outcomes did not convince me. I focus on the bars showing statistically significant correlations (blue or red coloured bars). It is clear that the fraction of the basin presenting high correlations lasts approximately for a time span equal to the time scale of the SPI: more or less 4 weeks when I use SPI1, more or less 13 weeks when I use SPI12 and so on. I'm not convinced that this is not simply due to time autocorrelation of the pairs SPIn(t), SPIn(t+n) and not to real physical processes as proposed in the discussions. Please, clarify this point as it is very important"

Regarding the concerns of the reviewer about the "potential autocorrelation" of the ETDI - SPI at 13, 26 and 52 weeks of aggregation, firstly, its mainly caused by the effect of the period of aggregation, and secondly, the purpose was in fact to illustrate the impact of the period of aggregation used for SPIw-13, w26 and w52 indices (of monthly focus as SPI-3, -6 and12) instead of using the weekly-focused results of SPIw-4 (SPI-1) to elucidate the interactions.

Since the Section 5.1 in results and 6.1 in discussion focus on the advantage of the week scale, we show the range of results from SPI-4w as supportive evidence of the pertinence of using the week scale, less prone to aggregated outcomes, particularly for this type of interactions and geographical context, than the commonly used monthly scale. In fact, many any times the SPI-3 is used to debate anomalies in the atmospheric system, like when referred to meteorological drought. We argue, based on the studies focused on flash droughts and drought on semi-arid environments, that the scale of analysis should be weekly, even when referring to interactions between atmosphere and land surface like SPI-ETDI or SPI-SMDI. For this reason, discussion in section 6.1 based on Figures 3,4,5,6 was written in that sense:

*L429-433 "The clusters of moderate to high correlation between indices mostly occur within the first month preceding or following an anomaly (Figs. 3-7), particularly in the short to very short-term. Apart from the tendency of high correlations to peak and plunge in the interval of a few weeks, the information about its delay or precedence can only be observed when the weekly scale is adopted."*

The range of subplots of Figs. 3 to 6, as well as the ones of aggregation period of 1 week and 78 weeks attached below, show in fact that the aggregation period mostly impacts results when using aggregations over 3 months, as it is the case of SPI-3 (SPIw-13), both in timing and duration of the clusters of lags. Therefore, the panel of aggregation periods of Figs. 3-6 provided two subplots below and two subplots over this midpoint of the range of aggregation to show the sensitivity of results to the aggregation period.

The additional plots of SPIw-n – ETDIw attached below (SPIw-1 and SPIw-78) show that the clusters of lags $MT_1$ and $ST_2$, primarily of precedent lags, appear clearly separated in the first three subplots of SPIw-1 - ETDIw, SPIw-4 - ETDIw and SPIw-13 - ETDIw while they become merged and largely distorted by the increasing period of aggregation (i.e. SPIw-26 and -52 – ETDIw). The longer the period of aggregation, the more the clusters increase the magnitude of the correlation, the length of the cluster and the significance of the correlations. Additionally, we can observe that the shortest period of correlation, the SPIw-1 – ETDIw shows very fragmented signals of interactions which is logical when analyzing results at the weekly scale. However, the timing of the fragmented clusters of correlation match well those shown until aggregation periods of 13 weeks, so that we can say, results are consistent in between weekly and the firsts months scale of aggregation of the SPIw. In consequence, the two upper ETDIw - SPIw-n subplots of Figure 3 and 4 aimed to illustrate the range of temporal scales at which the interactions between rainfall and evapotranspiration anomalies are within the range of observability. The lower ones of SPIw-26 and SPI-52 – ETDIw alternatively illustrate the temporal scales (seasonal, annual) at which the interactions between SPI and ETDI cannot be further discerned. In the case of SPI-n – SMDI, again the range of lag suitable for the interpretation of SPIw-ETDIw remains consistent until the seasonal scale of aggregation of the SPI.

The range of subplots in figs 3-6, aimed to illustrate how the weekly scale (dominating the clustering configuration in between SPIw-1 and SPIw-13) is the most suitable for the interpretation of the short-term interactions. For this reason, the results described in Section 5 mostly referred to the short-term aggregation periods to interpret the clusters of lags, stating that the higher SPIw26 and SPIw52 results tend to dampen, merge and later the clustering of lags indicated by SPIw4 and SPIw13. We are open to include SPIw1 in Figs 3-6 for better illustration of our purpose, while we refrain ourselves to include it due to the reluctance of the scientific community to refer to aggregations of the SPI index below the monthly scale. Therefore, we have further clarified this scope of showing the sensitivity of results to the aggregation period in the description of results (lines L331-332, L350-351, L393-395) and discussion section (L437-L440).

[Figure]

**SPIw-n - SMDI**

[Figure]

Regarding the possibility of autocorrelation, once checked the series are stationary (by Augmented Dickey-Fuller (ADF) test) we tested the autocorrelation results of each drought index series. ETDI and SMDI mainly show autocorrelations in the range from 7 to 10 weeks of significant autocorrelation. The autocorrelation values of series of SPI, while depending on the period of aggregation, differ notably (2 weeks on SPIw-1, 5 weeks on SPIw-4, 10 weeks for SPIw-13, 18 on SPIw-26, 35 on SPIw-52, 40 on SPI-78, 45 on SPIw-104) from those of ETDI and SMDI, while the range of ones of importance (ACF>0.5) barely reach half of them. The increasing values of autocorrelation with the increasingly aggregated SPI are compatible with the effect of the moving average of SPI. The evaluation of partial autocorrelation is more informative. Partial autocorrelations of ETDI and SMDI show mostly two (to four) week-lags as significant. An AR(2) configuration can be explained as a combination of growing and decaying exponentials. We assume the first term causes the direct relation and the second one inverse relation, supports our interpretation that the interactions between indices have a dominant positive interaction limited in time by a secondary inverse interaction, which we define as the energy-limiting shift from high-energy conditions (evapotranspiration-mediated) to low-energy conditions (rainfall-inhibited). The autoregression terms at 4, 13, 26, 52 of SPIw-n again refer to remnants of the moving average of the aggregation period of SPIw-n.

Therefore, since the duration of the autocorrelation of ETDI, SMDI and ETDI differs (except for the combination SPIw13-ETDI or SMDI) the results of significant interactions commented based on SPI-ETDI or SPI-SMDI interactions remain consistent from the weekly to the below-seasonal scale (SPIw-13), which is the scale at which we underline the importance of identifying the interactions between indices. Therefore, we can say that despite the evident increasing lengthening of the impact of the moving average (aggregation period) on the significant clusters of correlation between indices in subplots c) and d) of Figs.3-6, the duration of the lags shown in Figs. 3-6 does seem to be defined by true interaction between the anomalies of the indices beyond autocorrelation artifacts.

[Figure]

Regarding the apparent mismatch of RS and LSM of causes already discussed in Section 6.3 we can further illustrate for the reviewer the impact of the input and LSM structure. The parametrization of the model assumes a semi-distributed approach by sub-basins of the catchment on which each subbasin is defined based on average values of land cover and soil characteristics of the ECOCLIMAP2 database in the subbasin. In consequence, the patchiness of LSM results due to the partial

aggregation of the input (see figure below), may cause the loss of spatial variability compared to the remote sensing results and induce the mismatch on RS-LSM results we see on Figs. 3-7.

[Figure]

4. "Line 162 "In order to fill the gaps … interpolation". Please, specify the methodology adopted to interpolate and the maximum time span interpolated (this may strongly affect the results if the original time series is very fragmented, or the missing data interval are long)"

The interpolation was applied pixel by pixel on a temporal basis. The maximum time span for the temporal interpolation fed from the last previous data within two weeks.  No spatio-temporal interpolation was applied. This clarification has been included in the same L162 of the manuscript.

5. "Equations 1 and 3. I would suggest indicating the median with an overbar, avoiding MWS"

Unlike other authors applying the ETDI and SMDI, we considered worthwhile keeping the notation shown by the authors of the indices SMDI and ETDI (Narasimhan and Srinivasan, 2005, AFM) expressed in the Equations 1 and 10 of their article to avoid confusion in the formula defining these indices.

6. "Equation 1. As written, the first equation is always positive and the second is always negative. Is it correct? Shouldn't it be the opposite?"

The notation is that of Narasimhan and Srinivasan (2005, AFM) and is correct. We understand the misunderstanding of the reviewer since both the ETDI and SMDI may require an example to interpret their meaning. This is the case for Equation 1 of the water stress anomaly formula fed by the water stress ratio. The water stress ratio (WS=(PET-AET)/PET) for an area experiencing AET close to that of the PET (i.e. under wet conditions, e.g. 75% of PET) generates a water stress of WS=0.25. Using this value in the first equation of Equation 1, when WS<=MWS (assuming MWS may be 0.5, maxWS=0.9, minWS=0.1), we get the WSA=(MSW0.5-WS0.25)/(MWS0.5-minWS0.1)*100=62.5. This is a positive WSA that may tend to keep ETDI in + values, indicating wet conditions. It is the water stress ratio of values in between 0 (dry) and 1 (wet) which mislead the interpretation of the sign.

---

## Author Comment (AC2)

**Response to reviewer RC1 (**reviewer in black Open Sans 11, **response in blue, Calibri,11)**

"The study addresses a drought analysis in the Ebro River basin in Spain by using remote sensing (RS) data of evapotranspiration from MOD16A2ET and soil moisture data from SMOS1km 20 as well as SURFEX-ISBA land-surface model (LSM) data to calculate the Evapotranspiration Deficit Index (ETDI) and the Soil Moisture Deficit Index (SMDI) for the period 2010-2017. Also, SAFRAN data are used to calculate the Standardized Precipitation Index (SPI) at different aggregation time scales. The drought indices are computed both at the monthly and weekly scales. In particular, the study investigates the mechanisms of drought propagation in the land-atmosphere system by analyzing the temporal lags between the drought indices to identify the synchronicity and memory of the anomalies between precipitation, evapotranspiration and soil moisture to interpret factors involved in drought onset.

Overall, the study is interesting and well written. A few comments follow.

1) In the Introduction, the bibliographic review of previous studies on the use of remote sensing products in drought analysis at the weekly scale focused on rainfed agriculture could be extended. I suggest referring to the following study and reference therein: https://doi.org/10.5194/nhess-20-471-2020:"

We thank the reviewer for the insights of the suggested article about the multiple studies tackling the development of drought indices and the use of remote sensing products in drought analysis. The text has been of great help to significantly reform the introduction to further explain and clarify the aim of the study in the context of previous works. The first paragraph is now divided in two expanding the reasoning for the use of specific indices and the need to address evapotranspiration and soil moisture as relevant variables:

*"Drought is a major natural hazard for the societies in semi-arid climates (Van Loon, 2015) which demands increasing levels of adaptation and resilience measure to guarantee water supply (Watts et al., 2012), particularly in water-stressed environments. Rain-fed agriculture (Tigkas and Tsakiris, 2015), and even the enduring natural vegetation are very exposed to drought, especially under climate change, which has long-lasting implications to the local environment (Gudmundsson et al., 2014). Knowing that complex interactions take place in the land-atmosphere system under drought, the traditional meteorological or hydrologic approach may overlook drought-relevant interactions between evapotranspiration and soil moisture (Teuling et al., 2013).*

*This explains why modern drought monitoring combines evapotranspiration, soil moisture and even vegetation anomalies to track drought status, such as the Objective Drought Indicator (OBDI) integrated in the U.S. Drought Monitor (Svoboda et al., 2002) or the Combined Drought Indicator within the framework of the European Drought Indicator Observatory (Sepulcre-Cantó et al, 2012). This approach is on the upward trend, since even parsimonious composite drought indices like the probabilistic precipitation vegetation index (PPVI) (Monteleone, Bonacorso and Martina, 2020) outperform the capabilities of common indices to characterize drought. Therefore, composite indices facilitate the characterization of drought from multiple perspectives (e.g. Meteorological, Hydrological or Agricultural) but can be impractical to explore the mechanisms of drought due to complex calculations or missing data. Even though long-term anomalies of rainfall, and other meteorological, hydrological or vegetation condition variables evapotranspiration and soil moisture are currently regularly monitored, evapotranspiration and soil moisture ones still face challenging monitoring. Not only the indirect nature of these variables' data but also their limited spatial and temporal availability limit the number of studies adopting them, even despite and their known to play a relevant role in the recurrence of drought and heat waves (Zampieri et al., 2009; Dasari et al., 2014), often short-term anomalies are overlooked, Provided that especially regarding interactions of evapotranspiration and soil moisture operate on short time scales (Teuling et al, 2018), there is need to address dedicated exploration of*

*their relevance on the evolution of drought at the shortest time scale available, which for the soil moisture and evapotranspiration data is currently the weekly scale."*

In the fourth and fifth paragraph we have expanded on the reasoning of using remote sensing products with special reference to previous experience on this matter: *"Space agencies have released multiple RS products in the last decades facilitating the distributed analysis of drought (AghaKouchak et al., 2015). Optical spectrometry of the atmospheric (rainfall, temperature, water vapor) and surface (vegetation reflectance) variables have often been the basis for distributed characterization of drought indicators. Surface vegetation indices such as the widespread NDVI (Liu and Kogan, 1996) pioneered on the application of RS data to assess the impacts of drought, but thereafter the increasing availability of RS data of multiple meteorological variables has increased its usage on drought indices (West et al., 2019), While even common indices like the SPI can now rely on RS data (Sahoo et al, 2015), the many advantages of RS data facilitate integrating multiple data sources into the increasingly operative composite drought indices for weekly drought monitoring (USDM, Svovoda et al., 2002; CDI, Sepulcre-Cantó et al., 2012) even below the weekly scale (Monteleone, Bonaccorso and Martina, 2020). Beyond precipitation, temperature and other directly observable meteorological variables, evapotranspiration and soil moisture represent components of the land-atmosphere system which are difficult to measure on the ground, and consequently suitable for the focus of RS. Recent years have seen a rise in the availability of RS-based evapotranspiration databases such as the global dataset included in GLEAM (Miralles et al., 2011; Martens et al, 2017) or the soil moisture global database CCI (Dorigo et al., 2017). However, the still coarse spatial resolution of these global datasets limit the use of these databases for regional scale analysis or processes understanding.*

*Fortunately, in parallel to the RS missions, the development of processing techniques has improved the applicability of RS -derived data products (Wagner et al., 2007). On this basis, there are soil moisture datasets of increasing high resolution available from the combination of passive microwave sensors such as those from SMOS and SMAP missions (Kerr et al., 2010; Entekhabi et al., 2010; respectively) and active microwave sensors such as ASCAT or Sentinel-1 (Bartalis et al., 2007; Hornacek et al., 2012; respectively). This is the case of the high-resolution soil moisture and evapotranspiration products SMOS1km (Merlin et al., 2013; Molero et al., 2016, Escorihuela and Quintana-Seguí., 2016; Escorihuela et al., 2018). Similarly, high-resolution RS evapotranspiration products such as and the MOD16A2 (Mu et al., 2013) used in this study are currently available. Therefore, it is worth exploring the capabilities and limitations of high-resolution RS data for drought monitoring at regional scale. Both remote sensing products represent components of the land-atmosphere system which are difficult to measure on the ground, particularly under extreme conditions such as drought (Miralles et al., 2019). To date, relatively few works have used satellite data for drought analysis in the IP (Vicente-Serrano, 2006; Scaini et al, 2015, Martínez-Fernández et al., 2016; Sánchez et al., 2016; Ribeiro et al., 2019), especially at the spatial and temporal resolution of this study (Pablos et al., 2019)."* With these additions we hope the introduction provides a better idea on the background of the discipline and the purpose of the study.

2) "The complex interactions between drought indices, investigated by means of a throughout correlation analysis, highlights feedbacks among the considered variables, with a preeminent role of evapotranspiration in the link between rainfall and soil moisture. The study is carried out by calculating the Pearson correlation coefficients between pair of series of the three drought indices at weekly scale, introducing lags from -104 weeks to +104 weeks. The use of the Pearson correlation coefficient implies that the underlying variables are normal distributed. This is true, by definition, for the SPI, but what about the ETDI and SMDI? In case the normal hypothesis is rejected, the Spearman rank correlation statistic must be used instead. "

We sincerely appreciate the warning that ETDI and SMDI might differ from the normal distribution of the SPI. We are aware this difference implies a tendency to over-represent extremes by ETDI and SMDI compared to the SPI. There are several reasons of our choice of these indices despite this disadvantage: The use of r

Spearman was initially adopted but the data-intensive calculations required for the large amount of data made it impractical. The analysis of lags using Pearson, which took weeks for each interaction (i.e. SPI-4-SMDI), provided results in an order of magnitude quicker than the Spearman ones.  To test the representativity of the results of r Pearson coefficient presented in the manuscript, we have compared the lags calculated by Pearson and Spearman coefficients using a subset of the data ( ~1/4 of the total 10129 pixels: 2580 pixels, for the full range of lags from -104 to +104 weeks, chosen over a representative area of the basin, the north central region in between Pre-Pyrenees and the Ebro Depression). In this way, after several weeks of reanalysis we can indicate the similarity between coefficients is remarkable.  Correlation between the r Pearson and r Spearman results for the subset showed higher than 0.9 in all subsets based on RS data and lower for the case of the LSM data: 0.37-0.79. Therefore, considering the limitations of using Pearson under non-normal conditions, the notably higher computational cost of adopting Spearman instead of Pearson for such a big dataset (computational time in the order of months compared to in the order of weeks), but also the absence of characteristics of the data enhancing the difference between the two methods such as the existence of outliers (Orth et al., 2015), we assume r Pearson can be considered a reliable alternative to Spearman rank coefficient to evaluate the characteristics of the lags of our interest: timing, duration and magnitude. The results of the comparison between r Pearson and r Spearman are illustrated below for the SPI13w-ETDIw (r Pearson in blue, r Spearman in orange).

RS SPIw13-ETDIw

[Figure]

LSM SPIw13-ETDIw

3) "Lag analysis show a remarkable disagreement between RS and LSM for the SMDI - SPI interactions, with SMDI obtained with the LSM showing substantially lower correlations than the ones of RS, while also differing in the timing of the clusters of correlation. The authors state that they expected that the LSM, as being simpler than reality, had stronger SPI – ETDI - SMDI correlations than the RS dataset and justify this result by the accumulation of uncertainties of modelling, inputs and LSM structure. I suggest the authors to better argue this point.  Why should the adopted LSM work only for SPI and ETDI? "

The model simulations are run offline, this is, the meteorological data forces the LSM, but the LSM does not affect the forcing data. Thus, the feedbacks are lost. Some of them remain implicitly, as the meteorological observations include the feedback, as observed. For instance, previous works have shown that soil moisture is further influenced by additional factors beyond rainfall, such as groundwater and redistribution of soil moisture depending on topography, which are underrepresented in the LSM. This was reported from multiple other works such as the articles indicated by the reviewer (Teuling et al., 2006; Samaniego et al., 2018) and is a matter of improvement in the current versions of the model.

Beyond the causes commented above and already discussed in Section 6.3 we can further illustrate for the reviewer the impact of the input and LSM structure. The parametrization of the model assumes a semi-distributed approach by sub-basins of the catchment on which each subbasin is defined based on average values of land cover and soil characteristics of the ECOCLIMAP2 database in the subbasin. In consequence, the patchiness of LSM results due to the partial aggregation of the input (see figure below), may cause the loss of spatial variability compared to the remote sensing results and induce the mismatch on RS-LSM results we see on Figs. 3-7.

[Figure]

3b) Furthermore, gridded soil moisture datasets are available at the global and European scale (see for instance the Copernicus climate data service). These datasets can be a valid alternative to the LSM soil moisture data and deserve some references in the study.

The reviewer is right when indicating the many virtues of gridded soil moisture datasets available at multiple scales to further compare the performance of land surface models and remote sensing data in the characterization of soil moisture. Data from assimilated reanalysis models like ERA5 [0.1x0.1º], CCI [0.25x0.25º], LISFLOOD [5x5 km], to mention some, are available at spatial resolution over or nearly an order of magnitude coarser than the ones required for this study.

The reviewer must bear in mind, neither observational nor grided databases of comparable resolution of soil moisture and evapotranspiration are available in the area. Alternatives such as SMAP1 have very short series available (2015-onwards) despite their similar capabilities to SMOS1km (Dari et al., 2021).Previous studies have validated SMOS1km in the area of study *(Merlin et al., 2013, RSE)* [1x 1km resolution] or in nearby areas for MOD16A2 (Pasquato et al. 2015; Sanchez-Ruiz et al., 2016; García-Llamas et al., 2019).

Alternative coarser databases of similar remote sensing, reanalysis or modelling sources may incur in similar nature of the sources of uncertainties and consequently may incur in similar uncertainties to the evapotranspiration (MOD16A2ET) and soil moisture (SMOS1km) databases used in this study.

The use of the LSM SURFEX-ISBA in comparison to remote sensing data is a main aim of the study. Despite the limitations of this LSM, as offline model, it has been validated in the area before and successfully reported to provide useful evaluation of water resources (Escorihuela and Quintana-Seguí, 2016; Quintana-Seguí and Barella-Ortiz, 2020). Also in nearby areas like Portugal and France, the LSM SURFEX-ISBA has repeatedly shown great capability to simulate land-surface processes (Nogueira et al., 2020; Le Moigne et al., 2020). The comparison aims to detect the factors impacting the model performance to further improve the capabilities of the model.

4) With reference to the feedback mechanisms depicted in Figure 8, the Discussion can be enriched by a comparison with previous studies investigating the same mechanism, such as:

- In section 6.2 where the feedback mechanisms are discussed we considered valuable expanding (*in italics*) the clarification about the reason that supports defining the distinct mode of evolution of anomalies under high or low energy characteristics in the Mediterranean climate:

*"…compared to the prevalence of local soil moisture recycling common in more continental areas of Europe (Bisselink and Dolman, 2008). The advective explanation is supported by the contrast between the few weeks of precedent influence of soil moisture on rainfall we observe in the Ebro basin and the up to 250 days of precedent influence of continental areas prone to soil moisture recycling (Rowntree and Bolton, 1983; Bisselink and Dolman, 2008). Some studies focused on continental climates of relevant summer rainfall described the implications of the alteration of the recycling due to soil moisture depletion during heatwaves and drought (Rasmijn et al., 2018). In the Iberian Peninsula, though, due to the Mediterranean climate characterized by the lack of summer rainfall, soil moisture annually reaches such low levels that we can expect annual summer mechanisms of dry weather in the near atmosphere. Differences between areas where soil moisture plays a role, like central Europe, and areas where soil moisture is unable to control the evolution of the system under high-energy conditions, like the Iberian Peninsula, have been reported before in the Mediterranean-like western Australia in comparison to eastern Australia (Herold, Kala and Alexander, 2016)."*

- While regarding the mechanisms of the feedbacks, we appreciate the proposal of the reviewer about enriching the discussion, we find it appropriate and has been widely clarified and expanded accordingly partly to the suggested articles and some others into the text of section 6.2.

5) Please check Figure 7 since you wrote the same thing for both negative and positive lags.

Thanks for pinpointing the mistake in labels in the Figure 7 and when describing the lags in the caption "for the -104 leading and for the +104 lagged time steps of SMDIw in reference to the  ETDIw" in Line 799. It has been corrected

---

## Author Response (AR2)

**Response to corrections from reviewer 1:**

"I appreciate the authors' effort to improve the manuscript based on my previous suggestions. Overall, I think that the manuscript has been enhanced significantly. I only invite the authors to carefully read the text to perform an in-depth spell check and rephrase some of the new sentences to improve the quality of the language.
Some examples of sentences that need to be rewritten are at LL 48-52, LL 86-91 and LL 123-126."

Multiple paragraphs and sections have corrections according to the indications of Reviewer 1 to improve the clarity of the text, with special attention to rephrasing sentences referring too many results or information. Therefore, line numbers (with all revision visible) in L40-60, L93-111, L138-141, L288-296, L343-356, L370-385 and especially in the discussion section L488-515, L518-524, L533-559, L600-653.

Similarly, we checked spelling of the entire text carefully, and thank the reviewer for the suggestions.

**Response to corrections from reviewer 2:**

"I greatly appreciated the rigorous and deep work done by the authors to address the comments and remarks from me (but also from the other reviewer).
I'm not fully convinced yet by the discussion on the time lags, although I agree that the weekly time scale can be more informative to study the time dependency of the atmosphere-soil interactions.
However, I think that the representativeness of drought indexes, mainly when one wants to describe the time chain of atmosphere-soil processes, is an open research question. From this perspective, I think that the quality of the manuscript is excellent and even if I'm not fully convinced, it can support and feed the scientific discussion on a sound basis. At the end of the day, the validity of this kind of indexes, as well as of the adopted methodologies, will be measured on the ground of their efficiency and usefulness in a management framework. Therefore, from my side, congrats to the authors"

We thank the reviewer for the understanding of the value of the article despite the controversial approach. We understand the concerns of the reviewer on the representativeness of drought indices and the suitability of the methodology. In order to clarify that the article approach to the aims of highlighting the analysis of interactions as keystone of drought analysis is just one among the multiple possibilities to address the problem, we have included a specific paragraph in section 6.2. We state that this is one of the multiple approaches possible to the task, and that both drought indices and the methodology may have alternatives. Nonetheless, we conceive the article at least an interesting assessment of the mechanisms of drought with common tools. An investigation aimed to inform about the importance of scales in drought studies, to exhibit the power of recent datasets for drought analysis, and to underline the convenience of exploring the anomalies of specific key variables and their interactions to understand drought processes.